## RESEARCH ARTICLE

# LAMP1 and LAMP2A localise to axonal organelles with distinct motility dynamics and partially overlapping molecular signatures in human neurons

Reem Abouward[1,2,3,*], Alya Masoud Abdelhafid[1,2], Oscar G. Wilkins[1,4], Song-Yi Lee[5,6,7], Fairouz Ibrahim[4], Mark Skehel[4], Alice Ting[7,8,9,10], Nicol Birsa[1,4], Jernej Ule[3,4] and Giampietro Schiavo[1,2,‡]

## ABSTRACT

LAMP1 and LAMP2A (an isoform of LAMP2) are abundant proteins of late endosomal/lysosomal compartments that are often used interchangeably to label what is assumed to be the same organelle population, potentially obscuring distinct physiological roles. Here, we characterised the axonal transport dynamics of LAMP1- and LAMP2A-positive compartments in human induced pluripotent stem cell (hiPSC)-derived cortical neurons. We found that LAMP1-positive organelles move slower in the retrograde direction, pause more frequently, and display a broader anterograde velocity distribution than LAMP2A-positive vesicles, indicating distinct trafficking behaviours. Co-transport analysis revealed that ~65% of motile LAMP1-positive organelles carry LAMP2A, and vice versa, with higher co-transport in the retrograde direction. To explore molecular differences underlying these behaviours, we performed proximity labelling using full-length LAMP1 or LAMP2A fused to the light-activated biotin ligase LOV-Turbo. This approach revealed largely overlapping interactomes, with LAMP2A-associated proteins forming a subset of the LAMP1 interactome and showing an enrichment for synaptic vesicle-related proteins. We further validated ZFYVE16 as a novel interactor of both compartments. Together, our findings indicate that LAMP1- and LAMP2A-positive organelles share overlapping molecular identities but represent functionally distinct axonal populations with divergent transport dynamics.

**KEY WORDS: Axonal transport, Proximity labelling, Lysosomes, Endosomes, Synaptic vesicles**

[1]Department of Neuromuscular Diseases and UCL Queen Square Motor Neuron Disease Centre, UCL Queen Square Institute of Neurology, University College London, London WC1N 3BG, UK. [2]UK Dementia Research Institute, University College London, London WC1E 6BT, UK. [3]UK Dementia Research Institute, King's College London, London SE5 9RX, UK. [4]The Francis Crick Institute, London NW1 1AT, UK. [5]Department of New Biology, DGIST, Daegu 42988, Republic of Korea. [6]New Biology Research Center, DGIST, Daegu 42988, Republic of Korea. [7]Department of Genetics, Stanford University, Stanford, CA 94305, USA. [8]Department of Biology, Stanford University, Stanford, CA 94305, USA. [9]Department of Chemistry, Stanford University, Stanford, CA 94305, USA. [10]Chan Zuckerberg Biohub, San Francisco, CA 94158, USA.
*Present address: The Francis Crick Institute, London NW1 1AT, UK

‡Author for correspondence (giampietro.schiavo@ucl.ac.uk)

R.A., 0000-0001-9548-2599; A.M.A., 0009-0006-8917-2387; O.G.W., 0000-0002-3334-0568; S.-Y.L., 0000-0001-9091-8242; F.I., 0009-0007-1746-1339; M.S., 0000-0002-2432-0901; A.T., 0000-0002-8277-5226; N.B., 0000-0002-8271-540X; J.U., 0000-0002-2452-4277; G.S., 0000-0002-4319-8745

## INTRODUCTION

The endolysosomal pathway is a dynamic network of membrane-bound organelles required for several important cellular processes, including growth factor signalling, trafficking of proteins, lipids and RNA granules, and the regulation of protein translation and degradation (Cosker and Segal, 2014; Villarroel-Campos et al., 2018; Kuijpers et al., 2021; Vargas et al., 2022; De Pace et al., 2024). This pathway is formed by early endosomes (EEs), late endosomes (LEs), multivesicular bodies and lysosomes. However, these organelles undergo multiple fusion and fission events, maturing from one type to another, making the distinction between them rather blurred (Huotari and Helenius, 2011; Platta and Stenmark, 2011).

EEs are the first element of this complex network, sorting material endocytosed at the plasma membrane (PM) further along the pathway or recycling it back to the PM via recycling endosomes (Maxfield and McGraw, 2004). EEs mature into LEs, which are more acidic and highly motile, transporting internalised material towards the perinuclear region, where they eventually fuse with lysosomes (Gruenberg and Stenmark, 2004). Within their highly acidic lumen (pH 4.5–5.0), lysosomes house a wide range of acid hydrolases and lipases (Saftig, 2007), which are crucial for degrading endocytosed cargoes or cytoplasmic content, such as protein aggregates and damaged organelles, via autophagy (Perera and Zoncu, 2016; Galluzzi et al., 2017). Additionally, lysosomes function as signalling hubs, regulating processes such as nutrient sensing, $Ca^{2+}$ signalling and plasma membrane repair (Ballabio and Bonifacino, 2020; Settembre and Perera, 2024).

In neurons, the distribution of endolysosomal organelles is polarised, with lysosomes enriched in the soma and EEs in neurites, thus forming a gradient of organelles with increasing luminal acidity closer to the soma (Kulkarni and Maday, 2018). This means endosomal cargo requires transport over long distances to reach mature lysosomes for degradation, presenting unique challenges for neurons due to the length of their axons and complexity of dendritic arborisations. Therefore, it comes as no surprise that disruptions in the endolysosomal pathway are strongly associated with neurodegenerative disorders (NDs) (Sharma et al., 2018; Roney et al., 2022). In particular, lysosomal dysfunction underlies and/or exacerbates many NDs, either directly due to mutations in lysosomal proteins and hydrolases, or secondary to impairments in lysosomal integrity and/or autophagy (Wallings et al., 2019; Parenti et al., 2021). Moreover, altered motility of endolysosomal organelles, especially of their axonal transport, is an early feature of several NDs and has been causally linked to their pathogenesis (Sleigh et al., 2019; Roney et al., 2022). Given that dysfunction of the endolysosomal pathway is a hallmark of neurodegeneration, a detailed characterisation of lysosomal composition, heterogeneity and dynamics in neurons is crucial.

Two of the most used late endosomal/lysosomal markers are LAMP1 and LAMP2. These are type-1 transmembrane proteins with heavily glycosylated luminal domains, together accounting for ~50% of the protein content of lysosomal membranes (Eskelinen et al., 2003; Wilke et al., 2012). LAMP2 is the only member of the LAMP family of proteins to undergo alternative splicing, producing three distinct isoforms (LAMP2A, LAMP2B and LAMP2C) with different transmembrane and cytoplasmic regions, and tissue-specific expression patterns (Hatem et al., 1995; Furuta et al., 1999). Amongst them, LAMP2A is the best studied due to its role in chaperone-mediated autophagy (CMA), whereby proteins carrying a unique targeting motif are specifically delivered by chaperones to lysosomes for degradation (Kaushik and Cuervo, 2012).

LAMP1 and LAMP2 are similar in size and structure, sharing ~37% sequence similarity (Fukuda, 1991). Mice lacking both LAMP1 and LAMP2 die during embryonic development, and their fibroblasts show a buildup of autophagosomes and cholesterol in endolysosomal compartments, suggesting that LAMP1 and LAMP2 are required for lysosomal maturation and cholesterol processing (Eskelinen et al., 2004; Eskelinen, 2006; Huynh et al., 2007; Schneede et al., 2011). In contrast, LAMP1-knockout mice are normal with only mild brain astrogliosis (Andrejewski et al., 1999). LAMP2-knockout mice, however, show a more severe phenotype with a 50% mortality rate between postnatal days 20 and 40, due to skeletal muscle degeneration and reduced heart contractility (Tanaka et al., 2000). Consistent with this, variants in human *LAMP2* cause Danon's disease, a lysosomal storage disorder characterised by fatal cardiomyopathy, vacuolar myopathy and cognitive impairments (Nishino et al., 2000). Despite these functional differences, LAMP1 and LAMP2 continue to be interchangeably used to label organelles thought to be lysosomal in nature.

Over the past decade, studies have revealed that neuronal LAMP1 and LAMP2 localise to a diverse pool of endolysosomal organelles, many of which lack classical lysosomal features. In particular, Cheng et al. (2018a,b), have shown that only ~30% of LAMP-positive compartments exhibit degradative capacity in mouse dorsal root ganglion neurons, underscoring the functional heterogeneity of the endolysosomal system in neurons. Moreover, Grochowska et al. (2023) reported only ~40% colocalisation between LAMP1 and LAMP2 in mouse hippocampal neurons and demonstrated that dendritic vesicles carrying LAMP2A and LAMP2B (but not LAMP1) undergo activity-dependent membrane fusion to release CMA-targeted proteins into the extracellular space. These findings show that LAMP1 is not a reliable lysosomal marker (Cheng et al., 2018a,b) and suggest functional differences between dendritic LAMP1- and LAMP2-positive compartments (Grochowska et al., 2023). Consequently, the interchangeable use of LAMP1 and LAMP2 to label what is assumed to be the same endocytic pool of organelles might hinder functional classification of endolysosomal heterogeneity, thus hampering the correct interpretation of research findings and our understanding of NDs. Therefore, there is an urgent need to better define the molecular and functional determinants of LAMP1- and LAMP2A-positive organelles in neurons, particularly in axons where the transport of organelles is tightly linked to neuronal health (Villarroel-Campos et al., 2018; Sleigh et al., 2019).

In this work, we aim to characterise LAMP1- and LAMP2A-positive organelles in human induced pluripotent stem cell (hiPSC)-derived cortical neurons, focusing on LAMP2A due to the wealth of published information on its specific functions as compared to other isoforms. We used live imaging in microfluidic chambers and a novel proximity labelling approach, to compare the dynamic properties of these organelles in axons and uncover the interactors of LAMP1 and LAMP2A in human neurons.

## RESULTS

### LAMP1- and LAMP2A-positive organelles have distinct transport dynamics in axons

To study the axonal transport of LAMP1- and LAMP2A-positive organelles, we used microfluidic chambers (MFCs), an established system for separating somatodendritic and axonal compartments (Restani et al., 2012; Panzi et al., 2023). MFCs are made of two chambers connected by microgrooves (10 μm wide and 500 μm long) which due to their size, selectively allow the passage of axons, isolating them physically and fluidically from somas seeded in the 'somatic' compartment. We used hiPSCs genetically engineered to express neurogenin-2 (*NGN2*) under the control of a doxycycline (DOX)-inducible promoter, which allows their direct differentiation into cortical neurons (I3Ns) in a rapid, efficient and scalable manner using an established two-step protocol (Fernandopulle et al., 2018).

I3Ns were seeded into the 'somatic' chamber of MFCs and transduced at day *in vitro* (DIV) 10, with EGFP or mScarlet, fused to the cytoplasmic domains of LAMP1 or LAMP2A, respectively. Neuron-specific low expression levels of the two fusion proteins were ensured by a human synapsin-1 (*SYN*) promoter. At DIV13, live-imaging of the transport of LAMP1- and LAMP2A-positive organelles was carried out at the distal end of the microgrooves (axonal side). Data were anonymised and semi-manually analysed using Trackmate (Tinevez et al., 2017), capturing information on the displacement and speed of the moving compartments (Fig. 1A). Stationary organelles were rarely observed and were excluded from subsequent analysis. For terminally pausing organelles, a fixed number of frames was included so that pausing contributed to the average track velocity, while enabling consistent comparison of organelle movement prior to pausing.

Representative kymographs (Fig. 1B) show that LAMP1- and LAMP2A-positive organelles are highly motile, displaying diverse motion dynamics. To quantify their transport, the cumulative total displacement of each fluorescent spot was plotted over time for LAMP1 and LAMP2A (Fig. 1C). These plots reveal that for both populations, transport is bidirectional and varied, with puncta showing fast processive movements as well as slower motion modalities characterised by frequent pausing, and direction reversals. Notably, LAMP2A-positive organelles displayed more sustained, processive movement with fewer pauses than their LAMP1 counterparts. This observation prompted us to investigate the relative speed distribution of LAMP1- and LAMP2A-positive organelles (Fig. 1D). Although both showed a broad range of velocities in either the anterograde or retrograde directions, anterograde movement was generally faster. However, LAMP1-positive organelles had a wider anterograde speed distribution with seemingly two subpopulations of organelles, a slow and a fast one, compared to LAMP2A. Additionally, LAMP2A-positive compartments exhibited a distribution of retrograde movements shifted towards higher speeds compared to LAMP1 (Fig. 1D). Mean transport speeds per replicate are shown in Fig. 1E. LAMP1-positive organelles moved at 3.45±0.13 μm/s (mean±s.d.) anterogradely and 1.74±0.12 μm/s retrogradely, whereas LAMP2A-positive vesicles exhibited a comparable mean anterograde speed (3.77±0.33 μm/s) and a significantly faster retrograde speed (2.44±0.25 μm/s).

To investigate the degree of co-transport of LAMP1 and LAMP2A, I3Ns were co-transduced with LAMP1–EGFP and LAMP2A–mScarlet and axonal transport recorded using the same experimental setup. Additionally, we switched to a spinning-disk confocal

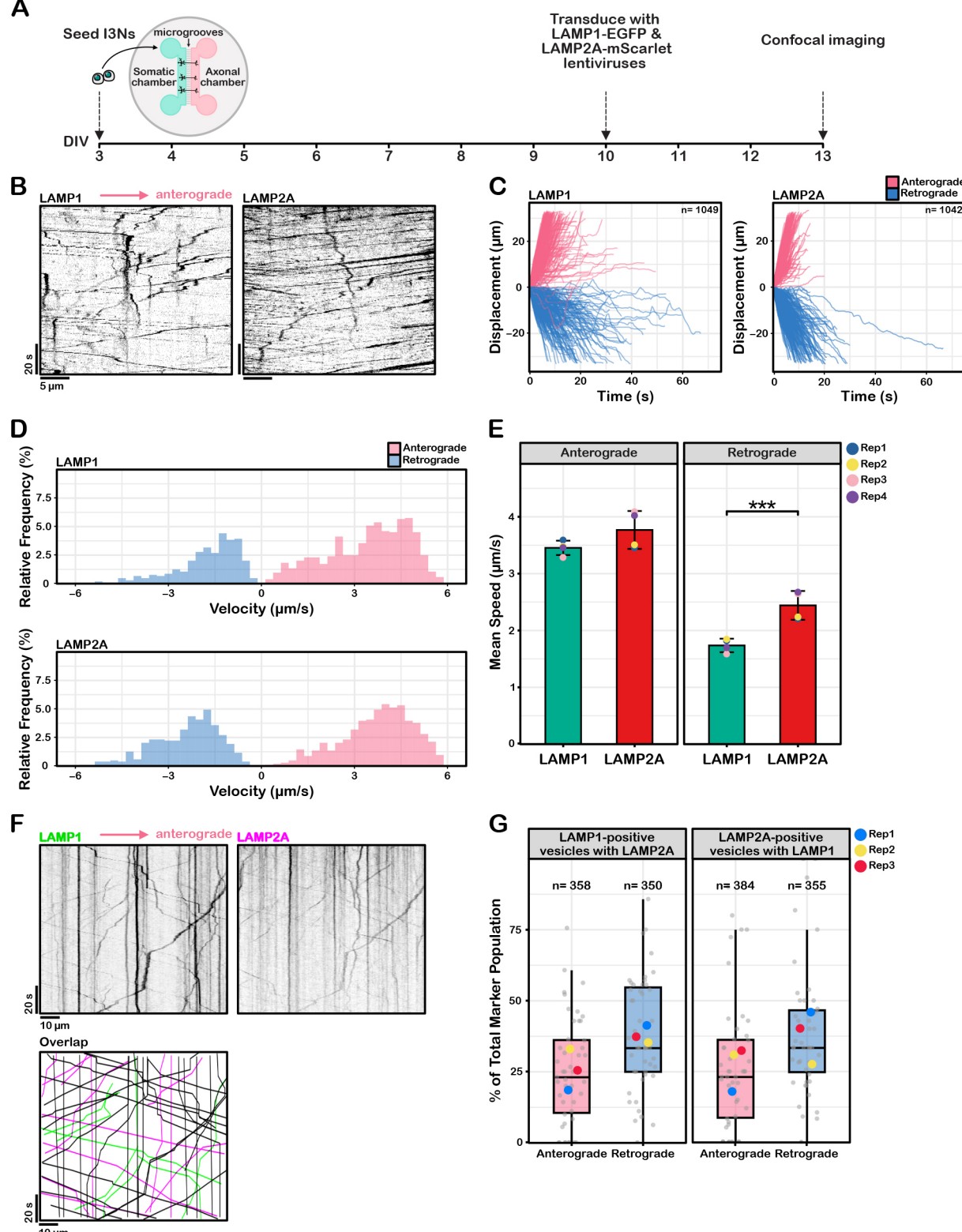

**Fig. 1.** See next page for legend.

microscope, instead of a laser-scanning system used for the above experiments, to enable imaging over a larger field of view with minimal laser-induced toxicity and bleed-through. Kymographs were generated over a 100 µm axonal segment in axons containing both LAMP1- and LAMP2A-positive puncta, and the proportion of colocalised motile tracks was quantified in each direction (Fig. 1F) across 10–17 axons per replicate with ∼1000 tracks analysed in total. For each direction, the percentage of colocalised tracks was calculated relative to the total number of motile tracks positive for either LAMP1 or LAMP2A within each kymograph (i.e. axon) per replicate, and

**Fig. 1. Transport dynamics of LAMP1- and LAMP2A-positive organelles.** (A) Experimental design and timeline for live-tracking organelle transport in axons. DIV3-differentiated I3Ns were seeded into one side of the MFCs. At DIV10, I3Ns were transduced with LAMP1–EGFP and/or LAMP2A–mScarlet, and at DIV13, live-imaging of the distal end of the microgrooves (~500 μm away from the soma) was carried out at 2 frames/s for 2 min. (B) Representative kymographs for LAMP1–EGFP and LAMP2A–mScarlet transport. Movement to the right (indicated by the arrow) is anterograde. (C) Cumulative displacement graphs of LAMP1-positive (left) and LAMP2A-positive (right) organelles over the recording period. Directionality was determined based on the displacement between the first and final location of an organelle. Organelles were tracked from four replicates and from at least three separate microgrooves per replicate. *n* is the total number of all organelles tracked. (D) Relative mean velocity distribution for LAMP1- (top) and LAMP2A-positive organelles (bottom). Each bin, 0.25 μm/s. Organelles that did not move were few and were not tracked. If organelles terminally paused after a period of motion, a maximum of 10 frames of pausing were recorded. (E) Mean±s.d. speed of LAMP1- and LAMP2A-positive organelles in the anterograde and retrograde directions. Mean±s.d. anterograde speed of LAMP1=3.4±0.126 μm/s, and LAMP2A=3.768±0.333 μm/s. Mean±s.d. retrograde speed of LAMP1=1.74±0.119 μm/s, and LAMP2A=2.441±0.253 μm/s. *P*-value=0.000871 (*** for *P*<0.001; Emmeans test using an error model from a mixed-effects linear model). Each dot is a biological replicate. Number of organelles tracked per replicate was comparable between LAMP1 and LAMP2A. For LAMP1, 252, 344, 194 and 259 organelles were tracked for Rep1 to Rep4, respectively. For LAMP2A, 297, 213, 307 and 225 organelles were tracked for Rep1 to Rep4, respectively. (F) Representative kymographs showing cotransport of LAMP1–EGFP and LAMP2A–mScarlet in axons of I3Ns co-transduced with both lentiviruses. Images were acquired at 2–10 frames/s for 2 min. Kymographs were generated and motile tracks manually traced for each channel. Tracks were overlapped to classify movements as single labelled (LAMP1-only, green; LAMP2A-only, magenta) or overlapping (black). (G) Quantification of colocalised motile tracks. For each direction, the percentage of overlapping tracks was calculated relative to the total number of motile tracks positive for the reference marker (LAMP1 or LAMP2A) within each axon (kymograph). Data were collected from 10–17 axons per replicate. *n* indicates the total number of motile organelles analysed per direction, including double-labelled tracks. Boxplots show the median and interquartile range (Q1–Q3), with whiskers representing minimum and maximum values; grey dots represent individual kymographs, and coloured dots indicate replicate means.

values were then averaged across replicates. This analysis revealed that ~64% of motile LAMP1-positive organelles carried LAMP2A, 25.7±7.2% (mean±s.d.) moving anterogradely and 38.1±2.7% moved retrogradely. Similarly, ~65% of motile LAMP2A-positive organelles carried LAMP1, split with 27.2±7.8% in the anterograde direction and 38.1±9.2% in the retrograde direction (Fig. 1G). Owing to inconsistencies with the acquisition frame rate during imaging, we were unable to accurately calculate speeds for the mobile tracks. However, using this spinning-disk setup, we observed more pausing events, which we similarly excluded from the analysis given that their quantification was challenging, with many organelles pausing at the same location, making it difficult to accurately quantify them.

To ensure that the observed differences were not attributable to the fluorescent tag, LAMP1–mScarlet was cloned into the same lentiviral backbone as LAMP1–EGFP. I3Ns were transduced with either construct, and axonal transport was monitored using a spinning-disk confocal microscope and tracked as before using Trackmate. Cumulative total displacement plots (Fig. S1A) showed that both LAMP1 fusion proteins displayed heterogeneous motility, including processive movement and frequent pausing, consistent with previous observations (Fig. 1). Similarly, relative speed distributions were comparable between the two constructs, with a greater proportion of slow-moving organelles in the retrograde direction (Fig. S1B). LAMP1–EGFP exhibited a mean anterograde

speed of 2.79±1.34 μm/s and a retrograde speed of 1.38±0.73 μm/s, whereas LAMP1–mScarlet showed a slightly lower anterograde speed of 2.16±1.19 μm/s and a similar retrograde speed of 1.40±0.72 μm/s (Fig. S1C).

In summary, our transport analysis shows that LAMP1- and LAMP2A-positive organelles have different motion dynamics, with LAMP1-positive organelles having a wider anterograde transport speed distribution, whereas LAMP2A-positive organelles exhibit more processive, faster transport, particularly in the retrograde direction. Additionally, ~65% of motile LAMP2A-positive organelles carry LAMP1, and vice versa, with a higher degree of co-transport occurring in the retrograde direction.

## Comparison of LAMP1 and LAMP2A interactomes

Given the differences we observed in axonal transport dynamics, we sought to further characterise the composition of the organelles carrying LAMP1 and LAMP2A by proximity labelling (PL). PL can be carried out in living cells to capture the proximal proteome of a protein of interest (Qin et al., 2021). In this approach, an enzyme (typically a biotin ligase or a peroxidase) is genetically fused to the protein of interest and under specific conditions (e.g. addition of biotin, or biotin-phenol and hydrogen peroxide), it catalyses the formation of small reactive molecules, which covalently label neighbouring proteins within a <20 nm radius. The chemical tag can then be used as an affinity handle for the purification and identification of the local protein environment by mass spectrometry (MS) (Qin et al., 2021). To avoid toxicity associated with the use of hydrogen peroxide, and for a more effective spatiotemporal control of biotinylation, we opted to use LOV-Turbo, a blue light-activated variant of the promiscuous biotin ligase TurboID (Lee et al., 2023). LOV-Turbo consists of TurboID fused to a light-oxygen-voltage (LOV) domain that suppresses ligase activity in the dark. Blue light induces a conformational change that activates TurboID allosterically, allowing proximity-dependent biotinylation in the presence of biotin and ATP (Fig. 2A) (Lee et al., 2023).

We generated fusion constructs of LOV-Turbo with LAMP1 (LAMP1–LOV) or LAMP2A (LAMP2A–LOV) designed to direct the ligase to the cytosol-facing surface of LAMP-positive organelles. In addition, LOV-Turbo was also fused to a nuclear export signal (NES), which targets it to the cytoplasm (NES-LOV), providing a spatial reference for cytoplasmic labelling (Cho et al., 2020). This control was important to distinguish organelle-associated proteins from non-specific cytoplasmic labelling, as the LAMP-positive organelles are highly dynamic. All constructs were driven by a *SYN* promoter (unless otherwise stated) and included a V5 tag as a linker between LOV-Turbo and the protein of interest, which facilitated subsequent validation by immunofluorescence and western blotting (Fig. 2B). We also designed and manufactured a light box using blue light-emitting diodes (LEDs) in house to reliably activate LOV-Turbo fusion proteins (Fig. S2A).

To test the light-dependent biotinylation of LOV-Turbo, DIV15 I3Ns were transduced with either LAMP1–LOV or NES-LOV and kept in the dark. At DIV18, 250 μM biotin was added to the medium and neurons were kept in the dark or exposed to blue light for 30 min pulsed at a 50% duty cycle (30 s on 30 s off) to minimise the risk of overheating. The cells were then stained with a V5-specific antibody, which displayed punctate distribution for LAMP1–LOV or diffuse cytoplasmic localisation for NES-LOV, with biotinylation only occurring when cells were exposed to blue light, as demonstrated by staining with fluorescent streptavidin (Fig. 2C). To verify this result by western blotting, HEK293T cells were transfected with a version of the NES-LOV construct identical to that used in Fig. 2B,

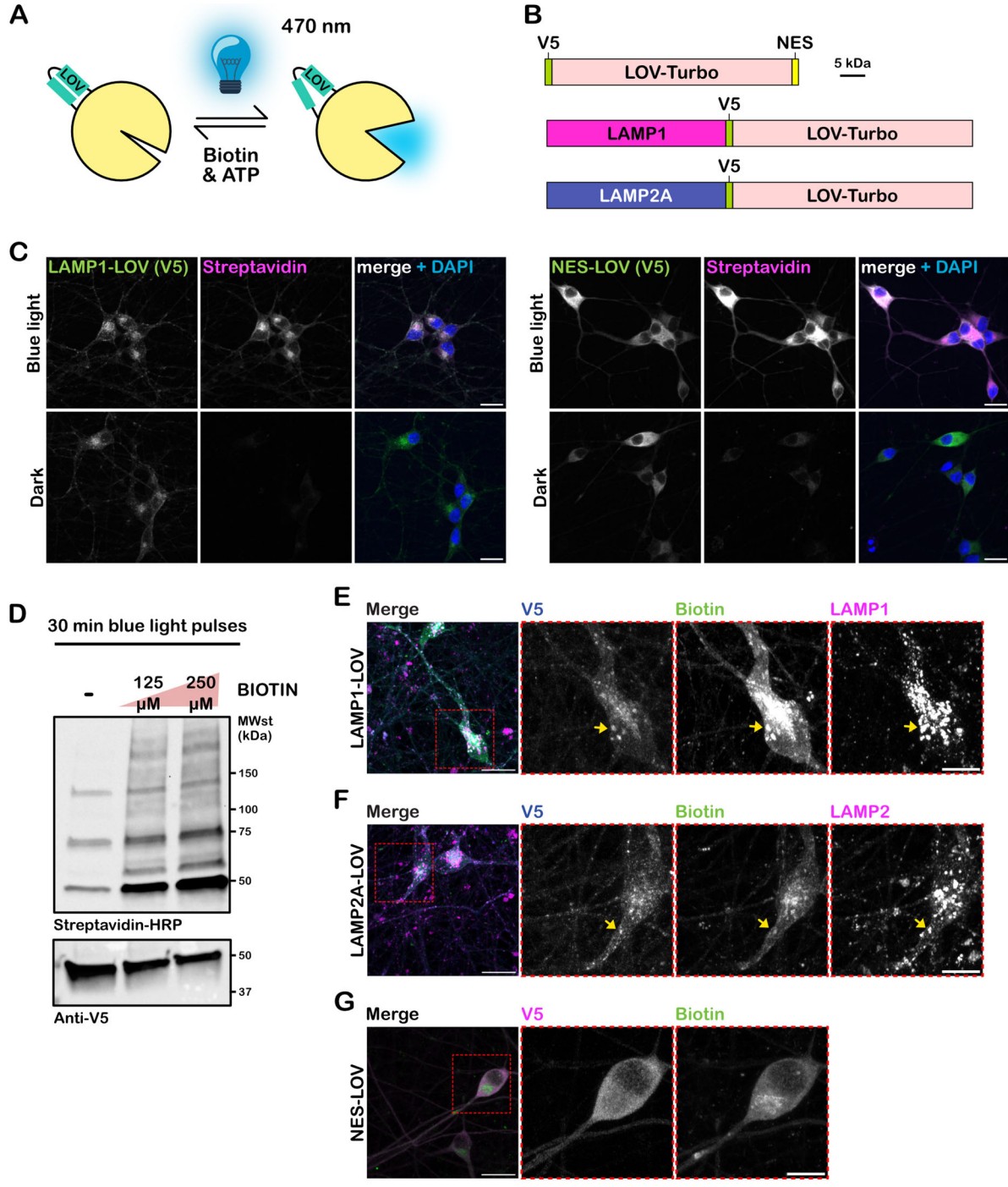

**Fig. 2. The localisation and biotinylation activity of LOV-Turbo fusion proteins.** (A) Schematic depicting the conformational changes of LOV-Turbo upon exposure to blue light (470 nm) leading to biotinylation if biotin and ATP are present (adapted from Lee et al., 2023). (B) LOV-Turbo fusion constructs generated in this work. LOV-Turbo is ~50 kDa. (C) LOV-Turbo regulation by blue light in I3Ns. I3Ns were transduced with LAMP1-LOV (top) or NES-LOV (bottom), fixed for imaging on DIV18 after three days of LOV-Turbo expression. Both were treated with 250 µM biotin on the day of the experiment and exposed to blue light pulses for 30 min. Scale bars: 15 µm. (D) Western blot showing the light-dependent activity of LOV-Turbo. I3Ns were transduced at DIV15 with NES-LOV and collected on DIV18, after treatment with two different concentrations of biotin and exposure to 30 min of blue light, as indicated. The blots were probed for biotinylation and protein expression by incubation with streptavidin-HRP and an anti-V5 antibody. Equal protein amounts were loaded across conditions. (E–G) Visualisation of LOV-Turbo fusion proteins by V5 staining for (E) LAMP1-LOV, (F) LAMP2A-LOV, showing colocalisation with LAMP1 or LAMP2, respectively and biotinylated proteins (visualised by streptavidin staining), magnified in inset. The yellow arrows point to sites of colocalisation. (G) NES-LOV (V5) is widespread across the cytoplasm, with similar localisation for biotinylated proteins, magnified in insets. Scale bars: 20 µm (main images, left panels), 10 µm (magnified views). Images in C–G are representative of a single experiment using Airyscan 2 imaging, validated by at least two repeats using non-Airyscan imaging.

but with a DOX-inducible promoter. The cells were kept in the dark and exposed to 30 min of blue light after DOX induction for 24 h. The western blots shown in Fig. S2B confirm the light-dependent activity of LOV-Turbo, as biotinylated protein smears were only visible when cells were exposed to blue light, whereas only endogenously biotinylated proteins were detected in the dark.

Similar results were observed in I3Ns (Fig. 2D). I3Ns were transduced with DOX-inducible NES–LOV and collected on DIV18, following 3 days of DOX treatment and exposed to blue light with no additional biotin, or upon addition to the medium of a final concentration of 125 or 250 µM of biotin. Fig. 2D shows increasing biotinylation with higher biotin concentrations, with only endogenously biotinylated proteins at ~50, ~75 and ~150 kDa detectable in the absence of additional biotin.

Having established that LOV-Turbo fusion proteins enable light-dependent biotinylation, we used high-resolution confocal imaging in I3Ns to confirm whether they colocalise with their endogenous counterparts and biotinylated proteins. Neurons were transduced and treated as described above, with all constructs regulated by the *SYN* promoter. Both LAMP1– and LAMP2A–LOV colocalised with LAMP1 and LAMP2, respectively, and with the biotinylated protein signal (Fig. 2E,F). In contrast, NES–LOV was distributed across the cytoplasm with similarly diffused biotinylation signal, albeit the latter also shows an enrichment in an unknown pool of perinuclear organelles (Fig. 2G).

We then proceeded to identify the LAMP interactomes using LAMP1–, LAMP2A– and NES–LOV samples and an untransduced (UT) control. Neurons were first transduced with the corresponding lentivirus on DIV15 and biotinylation carried out as previously described in four independent replicates per condition. Following streptavidin pulldown, western blotting of a small portion of the eluate showed that the biotinylation was highest in the LAMP1–LOV sample with most biotinylated proteins depleted from the flow through (Fig. S3A). Samples were processed using label-free data-dependent MS over two rounds, once with one run per condition and another with three technical runs per condition. Regrettably, a NES–LOV sample was lost during processing. Each condition yielded >1000 proteins (Fig. S3B). Principal component analysis (PCA) showed clear clustering by condition, with a batch effect between MS runs, reflecting the known variability associated with MS data acquisition (Karpievitch et al., 2012) (Fig. 3A).

To identify enriched proteins, we calculated $\log_2$ fold-changes of protein intensity in LAMP1–LOV and LAMP2A–LOV samples relative to both NES–LOV and UT controls. If a protein was found enriched in the UT condition in any comparison ($\log_2$ fold-change>1.5 and adjusted *P*-value<0.05), it was filtered out of the full dataset. Remaining proteins were taken forward for further analysis, comparing the LAMP1 and LAMP2A datasets to the NES control (Fig. 3B,C). In addition to enriched proteins identified through this analysis, 375 and 67 proteins were uniquely detected in the LAMP1–LOV and LAMP2A–LOV samples, respectively, and were not present in the NES–LOV control. These proteins were not imputed as part of the MSstats package used for MS analysis. The top ten most abundant proteins of this group are listed in Fig. 3B and C, ranked by decreasing intensity.

Proteins present in the LAMP1-LOV dataset include LAMP1 itself and several known functional constituents of lysosomes, such as BORC5, a subunit of the BORC complex regulating lysosome positioning within the cell, KIF1A, which regulates lysosomal transport, and the ATP6V0A1 subunit of the v-ATPase which is known to mediate lysosomal acidification (Ballabio and Bonifacino, 2020). Similar observations were made for the LAMP2A–LOV dataset, where LAMP2 itself is exclusively detected, together with NDRG4, a protein regulating brain derived neurotrophic factor (BDNF) in the brain (Yamamoto et al., 2011). Owing to their biological relevance, these proteins are included in the total LAMP1 and LAMP2A interactome dataset in Table S1.

Gene set enrichment analysis (GSEA) for the LAMP1–LOV dataset shows an enrichment of endosomal and lysosomal gene ontology (GO) terms (Fig. 3D). Compared to LAMP1–LOV, the LAMP2A–LOV dataset had fewer significantly enriched proteins over the cytoplasmic control (Table S1). This is possibly due to a lower expression level of LAMP2A–LOV and consequently less biotinylation, as suggested by enrichment ratios in the volcano plot (Fig. 3C), or because LAMP2A-interacting proteins are equally abundant in the cytosol. Nonetheless, GSEA for LAMP2A–LOV shows an enrichment in clathrin-coated vesicle-related GO terms (Fig. 3E). Interestingly, we found several synaptic proteins, such as SYN1, SV2A, SYT1 and SNAP25, in the interactomes of both LAMP1 and LAMP2A. However, in the LAMP2A–LOV dataset, synaptic proteins constituted a larger proportion of the total interactome, resulting in the 'synaptic vesicle membrane' GO term reaching the top ten enriched terms (Fig. 3E). Further analysis revealed that LAMP2A interacts with a subset of the total LAMP1 interactome (Fig. 3F), with the remaining LAMP1-specific interactors enriched for lysosomal and late endosomal GO terms (Fig. S3C).

In summary, we were able to capture comprehensive interactome datasets for LAMP1 and LAMP2A in human I3Ns, with GSEA highlighting that LAMPs interact with a diverse repertoire of endolysosomal proteins.

### LAMP1 and LAMP2A interactome validation

To validate our results, we selected two proteins with synaptic GO term localisation, synaptotagmin 1 (SYT1) and SNAP25. SYT1 is the main $Ca^{2+}$ sensor for synaptic vesicle exocytosis (Sudhof, 2004), whereas SNAP25 is a SNARE protein essential for synaptic vesicle fusion (Sudhof, 2004). I3Ns were fixed on DIV18 and stained for LAMP1 and/or LAMP2 (LAMP1/2) with SYT1 or SNAP25 and imaged using high-resolution confocal microscopy to test whether these synaptic proteins are spatially co-distributed with LAMP1 and LAMP2 (Fig. S4). Neurofilament heavy chain (NFH) staining was used to label axons. Intensity profile plots across LAMP1/2-positive organelles in NFH-positive projections, show partial colocalisation with small SYT1 and SNAP25 puncta decorating the surface of these organelles (Fig. S4). However, quantifying the degree of overlap of SYT1 and SNAP25 with LAMP1/2, was challenging due to the high abundance of these proteins in neurons.

To enable a more precise quantification, we employed a proximity ligation assay (PLA), which enables the detection of the co-distribution and localisation of two proteins within a ~40 nm radius. Co-distribution is marked by discrete fluorescent puncta on an otherwise negative background (Klaesson et al., 2018; George et al., 2022). For this analysis, we focused on SNX3, a member of the sorting nexin family mainly associated with early endosomes highly enriched in our dataset, and ZFYVE16 (also known as endofin), an early endosomal protein implicated in regulating membrane trafficking that was not previously known to interact with LAMP1/2, for validation. RAB7A, a master regulator of the endolysosomal pathway, was used as a positive control (Fig. 4). PLA was carried out in DIV18/19 I3Ns, and DAPI staining was used to quantify the total number of neurons. All proteins tested showed abundant PLA puncta with both LAMP1 and LAMP2 (Fig. 4A). Negligible signal was detected in the negative controls, whereby only one of the two primary antibodies was omitted (Fig. S5). Data were quantified using an automated analysis pipeline to segment and count the number of PLA puncta and assess DAPI signal, calculating the number of puncta per neuron (Fig. S6). Three

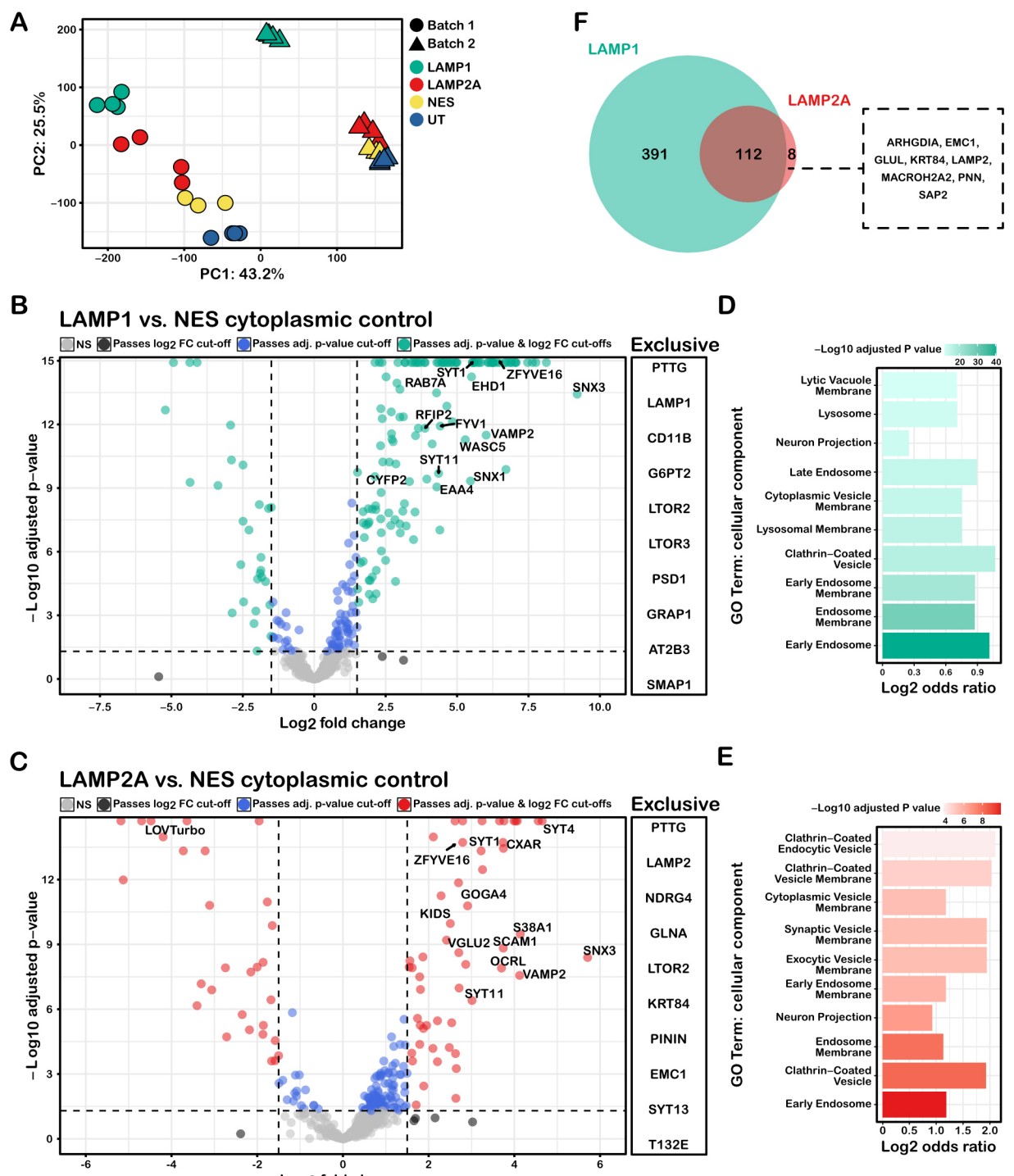

**Fig. 3. The interactomes of LAMP1 and LAMP2A in I3Ns.** (A) PCA using the top 500 most variable proteins. Each dot is a biological replicate, coloured by condition. All four conditions were run twice on a MS machine, named batch 1 and 2 in the figure, with batch two being an average of three technical runs. There were four replicates per condition, except for NES–LOV ($n$=3). (B,C) Volcano plots of the fold change in $\log_2$ protein intensities of (B) LAMP1–LOV and (C) LAMP2A–LOV over NES–LOV control, plotted against $-\log_{10}$ adjusted $P$-values (adj. $P$-value). The dotted lines mark the adj. $P$-value and $\log_2$ fold-change ($\log_2$ FC) cut-offs of 0.05 and 1.5, respectively. NS refers to non-significant proteins. Proteins at the top of the graph have infinite $P$-values and cannot be fit on the graph. Additionally, some proteins were only detected in the LAMP1– or LAMP2A–LOV conditions and not in the NES control. The top 10 of these are listed on the right of the volcano plot ranked by intensity. (D,E) GSEA showing the top 10 enriched terms (GO cellular component) for the proteins significantly enriched in the (D) LAMP1–LOV or (E) LAMP2A–LOV condition over the NES cytoplasmic control, including those detected exclusively in LAMP1– or LAMP2A–LOV datasets. All proteins identified in the MS were used as a background dataset for GSEA. (F) Overlap between LAMP1– and LAMP2A–LOV datasets.

biological replicates were performed per protein of interest, with a LAMP1- or LAMP2-only negative control included with each replicate for normalisation purposes. This analysis confirmed that SNX3, ZFYVE16 and RAB7A are in close proximity to LAMP1 and LAMP2, as indicated by the robust PLA signal compared to negative controls (Fig. 4B,C).

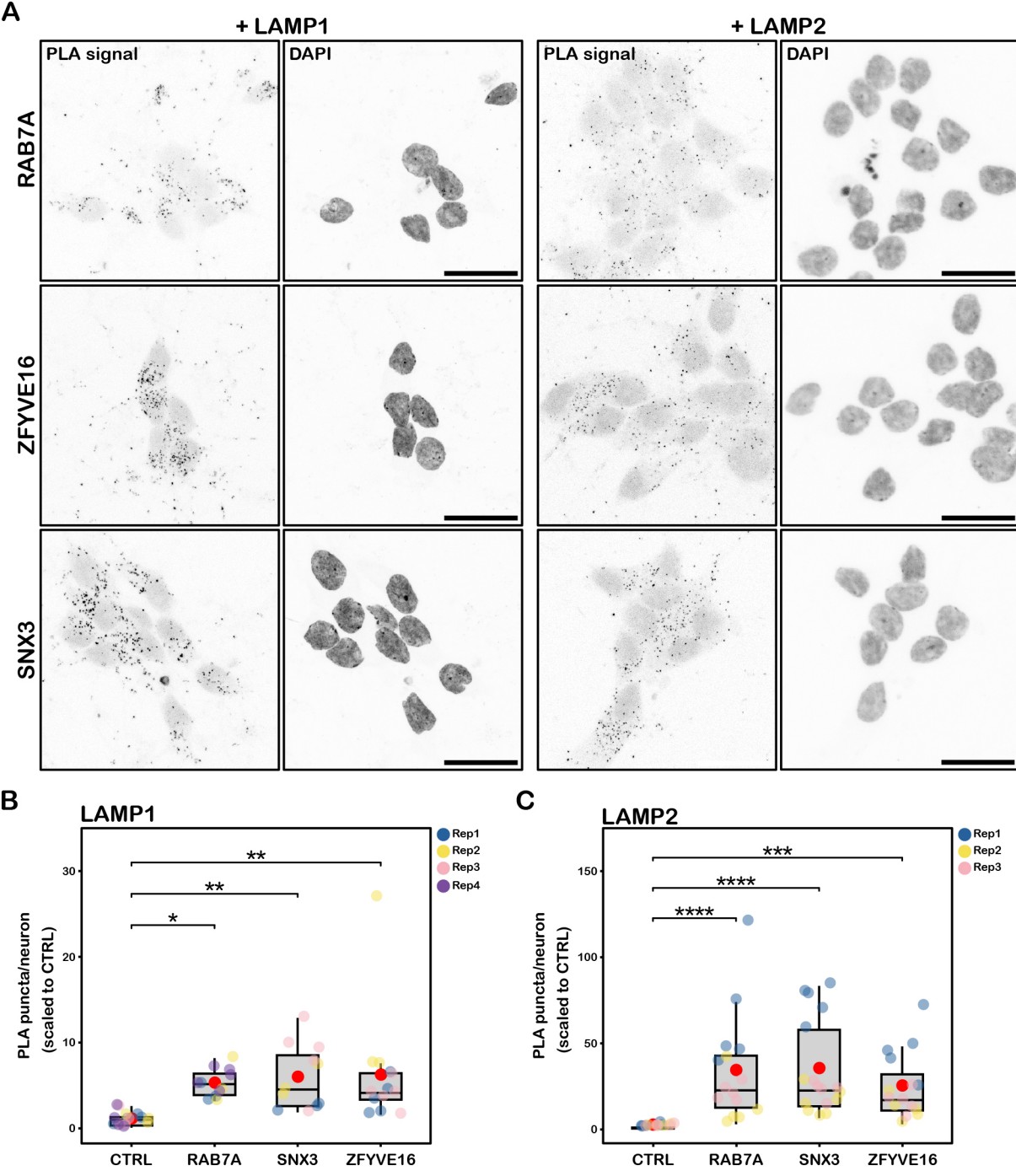

**Fig. 4. Validation of identified interactors of LAMP1 and LAMP2 via PLA.** (A) Representative PLA images for LAMP1 and LAMP2 to validate interactomes identified by LOV-Turbo biotinylation. The two top enriched proteins (ZFYVE16 and SNX3) were selected in addition to a positive control, RAB7A. DIV18 or 19 I3Ns were fixed and incubated with primary antibodies against RAB7A, ZFYVE16 and SNX3, either with anti-LAMP1 (left panel pairs) or anti-LAMP2 primary antibodies (right panel pairs). For all proteins, neurons were fixed and the PLA carried out simultaneously with their respective controls. All images were captured using the same settings as the controls. However, the images presented here have their brightness and/or contrast adjusted to enable better visualisation of the PLA puncta. Scale bars: 20 μm. (B,C) PLA quantification. The number of PLA puncta per neuron (nuclei count), was normalised to the average count in a LAMP1-only (B) or LAMP2A-only (C) primary antibody control (CTRL) per replicate. A total of 3–5 images were taken per replicate, with three replicates in total per protein of interest. Data are presented as boxplots with median and interquartile range (Q1–Q3), with whiskers representing minimum and maximum values; the red dot highlights the mean. *$P<0.05$; **$P<0.01$; ***$P<0.001$; ****$P<0.0001$ (adjusted $P$-value determined by an Emmeans test using error models from a mixed-effects linear model, and multiple testing correction using the FDR method of Benjamini and Hochberg).

In summary, we were able to confirm the results obtained by LOV-Turbo using imaging-based techniques as orthogonal validation approaches. For these validations, we focused on common interactors to ensure that the data were of sufficient quality to support future studies, as the number of proteins uniquely detected in the LAMP2A dataset was limited and might reflect experimental constraints.

## DISCUSSION

In this study, we characterised the transport dynamics and interactomes of LAMP1 and LAMP2A in hiPSC-derived cortical neurons. We found that the two proteins associate with a heterogeneous population of organelles with a wide range of velocities. We also demonstrated partial co-transport of LAMP1- and LAMP2A-positive organelles, with a greater proportion of co-transport observed in the retrograde direction. Additionally, PL with LOV-Turbo revealed interactions with a range of proteins, including with early endosomal and synaptic components.

Our data show that LAMP1-positive organelles exhibited faster anterograde speed (3.45±0.13 µm/s; mean±s.d.) compared to published *in vivo* speed in mouse thalamocortical axons (2.37 µm/s) (Nassal et al., 2022). This *in vivo* speed was itself higher than mean speeds measured *in vitro* in hiPSC-derived neurons (~1.5 µm/s) and rat hippocampal neurons (1.5–1.7 µm/s and 2 µm/s) (Boecker et al., 2020; De Pace et al., 2020). The retrograde LAMP1 speed (1.74±0.12 µm/s) was more comparable to that measured *in vivo* in mouse thalamocortical axons (1.48 µm/s) (Nassal et al., 2022). To our knowledge, the only published analysis for LAMP2A-posive organelles was performed in hippocampal rat neurons 1 and 4 h after LAMP2A release from the endoplasmic reticulum (ER) (Li et al., 2024). In this study, authors measured LAMP2A-positive organelle speeds between 1 and 1.5 µm/s in the anterograde direction, and 0.75–0.83 µm/s in the retrograde direction, being faster for newly released proteins. In contrast, we observed significantly higher speeds (3.77±0.33 µm/s anterograde; 2.44±0.25 µm/s retrograde). This discrepancy might stem from differences in methodology; previous studies relied on kymograph analysis of ~100–500 organelles, whereas we semi-manually tracked over 2000 organelles using TrackMate, allowing for the inclusion of small or dim puncta often overlooked in kymographs. Furthermore, Li et al. (2024) focused on proximal axons in rat neurons, whereas our measurements were taken in distal axons (>500 µm from the soma) of human iPSC-derived neurons.

We also found that retrograde transport is slower than anterograde transport for both LAMP1- and LAMP2A-positive organelles, consistent with published data (Boecker et al., 2020; Li et al., 2024). Interestingly, LAMP2A-positive organelles moved significantly faster in the retrograde direction compared to LAMP1-positive compartments, suggesting differences in motor protein recruitment and/or regulation. In the anterograde direction, LAMP1- and LAMP2A-positive organelles have similar mean speeds, with velocity histograms suggesting the presence of two distinct pools. The LAMP2A-positive population was more processive, with fewer slow-moving organelles, suggesting reduced pausing events or distinct regulatory mechanisms. Our observed differences were not attributable to the fluorescent tag, as LAMP1–mScarlet and LAMP1–EGFP exhibited comparable transport behaviours. Both constructs displayed similarly low retrograde speeds (~1.4 µm/s, comparable to ~1.7 µm/s measured previously), whereas LAMP1–EGFP showed a modestly higher anterograde speed (2.79±1.34 µm/s) than LAMP1-mScarlet (2.16±1.19 µm/s). These anterograde speeds were lower than those measured in our larger dataset comparing LAMP1 to LAMP2A (3.45 µm/s), which we attribute to the substantially smaller number of organelles analysed here (~100 versus >1000 previously), as reflected by the higher variance in velocity measurements.

Furthermore, we assessed the degree of colocalisation between the two proteins during axonal transport and found that of ~65% of all motile LAMP1-positive organelles (~25% moving anterogradely and ~38% moving retrogradely) also carried LAMP2A, with similar values of LAMP2A-positive vesicles with LAMP1. In a recent work in rat hippocampal neurons by Li et al. (2024), they measured a lower percentage of axonal LAMP2A-positive organelles co-transported anterogradely with LAMP1 compared to retrogradely, thus validating our findings. Moreover, they showed that acidic organelles in axons were mostly also positive for LAMP1, with gradual recruitment of LAMP2A overtime. Taken together with our motility data, this partial co-transport suggests distinct pools of both LAMP1- and LAMP2A-positive organelles, with functionally different motility behaviours, perhaps through differences in associated adaptor proteins facilitating differential motor recruitment (Cason and Holzbaur, 2022). These conclusions align with published data reporting the existence of subpopulations of LAMP-positive compartments (Cheng et al., 2018a; Kulkarni and Maday, 2018). A recent super-resolution study in non-neuronal cells identified eight lysosomal subtypes based on late endosomal/lysosomal protein distributions across LAMP1/2-positive organelles, although LAMP1 and LAMP2 were largely colocalised in this cell model (Bond et al., 2025). In neurons, however, LAMP1 and LAMP2 are localised to functionally distinct organelles in dendrites (Goo et al., 2017; Grochowska et al., 2023) and are sorted into distinct post-Golgi vesicles and trafficked separately into axons (Li et al., 2024). Moreover, newly synthesised LAMP2A-containing vesicles moved faster than more mature organelles in axons (Li et al., 2024), reflecting plasticity in organelle function over time. Further studies characterising organelle-associated CMA activity, pH and enzymatic content will help to functionally characterise the distinct pools of axonal LAMP1- or LAMP2A-positive organelles.

LOV-Turbo PL was then used to glean new insights on the molecular underpinnings of these transport differences. Through PL in mass culture I3Ns, we were able to capture several known interactors, such as KIF1A, BORC5 and LAMTORs (Farías et al., 2017), and several pre-synaptic proteins, such as SYN1, SV2A, SYT1 and SNAP25. Additionally, we verified the co-distribution of SYT1 and SNAP25 with LAMP1 and LAMP2 via confocal imaging. Recently, it has been shown that LAMP1/2 co-segregates with SYT1 in the Golgi and is co-transported into axons, independently validating our findings (Li et al., 2024). However, further studies are needed to clarify the percentage of LAMP2A-positive organelles that carry synaptic proteins and whether this differs from LAMP1-positive compartments. It is also unclear how these synaptic proteins distribute between organelles and whether these mechanisms are conserved across species, for example, whether a specialised type of organelle carries all presynaptic proteins, or if they are carried by different compartments based on specific molecular determinants (Roney et al., 2022).

The LAMP1-LOV dataset we generated was smaller than a published LAMP1-TurboID dataset (~1500 versus >2500 proteins identified, respectively) (Liao et al., 2019). However, we were able to enrich a higher percentage of significant proteins, despite applying stricter thresholds. Notably, we did not capture RNA granule interactors, such as FUS and G3BP1, which were previously linked to RNA hitchhiking and local translation in neurons (Cioni et al., 2019; Liao et al., 2019; De Pace et al., 2024). This discrepancy might be due to a combination of low, physiologically relevant LOV-Turbo expression and steric hindrance, allowing the detection of only the most proximal interactors of LAMP1 and LAMP2. In fact, both BORC5 and KIF1A, which were detected in the LAMP1 interactome, directly bind to the lysosomal surface (Pu et al., 2015; Guardia et al., 2016). RNA-binding proteins, such as G3BP2 and FUS, found associated with LAMP1-positive compartments in other studies (Liao et al., 2019; Li et al., 2024), are components of

RNA granules and might not be in direct contact with the surface of LAMP1-positive organelles. Nonetheless, we found several ribosomal proteins in our LAMP interactome, suggesting that LAMP-positive organelles could support local translation.

A possible limitation of our approach is that we did not stimulate and/or quantify CMA activity. The binding of client proteins to LAMP2A is a rate-limiting step of CMA and specifically differentiates this protein from other LAMP family members (Cuervo and Dice, 2000; Kaushik and Cuervo, 2018). Whereas the levels of CMA activity taking place in I3Ns is currently unknown, the LAMP2A interactor and essential CMA chaperone HSC70 (Agarraberes and Dice, 2001; Kaushik and Cuervo, 2018), was not detected in our study, suggesting a low level of CMA activity in our experimental conditions. Additionally, unlike LAMP1, LAMP2A is known to dynamically associate to cholesterol- and glycosphingolipid-rich microdomains in a process dependent on CMA activity (Kaushik et al., 2006). Thus, in the absence of CMA activation, LAMP2A might be sequestered in specific areas of the membrane, further restricting its interactions. Moreover, we relied on mass culture, in which the biotinylation signal might be saturated by somatic, mostly immobile, LAMP-positive organelles, unlike what is seen in the highly motile axonal carriers. Therefore, we were unable to identify a distinct set of proteins associating specifically with LAMP2A. Despite several attempts using various custom-made neuronal culturing devices, we were unable to obtain enough biotinylation signal from axonal preparations for a reliable MS analysis, thus leaving the identity of these axonal organelles a hanging question.

Overall, our findings support the view that LAMP1 and LAMP2A label distinct, yet partially overlapping organelle populations, with different motility and molecular signatures, and reinforce the idea that LAMPs are not specific markers for mature lysosomes in neurons (Cheng et al., 2018a; Li et al., 2024; Bond et al., 2025). Importantly, we showed that a combinatorial strategy adopting a variety of independent approaches is required to accurately compare the identity of organelles. Perhaps, new methodologies are needed to study sub-compartment organelle identity, potentially borrowing from recent advances in the single-cell proteomic field. Future work is needed to capture the dynamic changes to organelle identity under physiological stimuli and in the context of neurodegeneration and ageing. Notably, several key pathogenic proteins, including tau, α-synuclein and amyloid precursor protein, hijack pre-existing transport routes and organelle subtypes for propagation across neuronal circuits (Abeliovich and Gitler, 2016; Nixon, 2017; Evrard et al., 2018; Soares et al., 2021; Xie et al., 2022). Moreover, targeted manipulation of organelle identity is emerging as a promising therapeutic strategy for NDs (See et al., 2021 preprint; Hung et al., 2023). Therefore, a deeper understanding of the molecular and functional heterogeneity within the neuronal endolysosomal network is crucial for advancing both basic neuroscience and therapeutic development.

## MATERIALS AND METHODS
### Human iPSC and neuronal cultures
Cells were cultured in a humidified incubator at 37°C with a 5% $CO_2$ supply and routinely tested to ensure they were mycoplasma free. Experiments were carried out using neurons differentiated from hiPSCs (male; WTC11 background) with a DOX-inducible neurogenin-2 (NGN2) cassette inserted into a safe AAVS1 locus, and CAG promoter driven expression of catalytically-dead Cas9 fused to KRAB transcriptional repression domain (dCas9-KRAB), inserted into CLYBL intragenic safe harbour site (Wang et al., 2017; Fernandopulle et al., 2018; Tian et al., 2019). iPSCs were cultured on plates coated with Geltrex basement membrane matrix (Gibco) in mtseR Plus (STEMCELL Technologies) medium, following published

guidelines (Fernandopulle et al., 2018). For passaging, they were split using Versene (Gibco) with Accutase (Sigma-Aldrich) used prior to neuronal induction. For neuronal induction, iPSCs were switched to induction medium consisting of DMEM/F-12-GlutaMAX media (Gibco), 1× N2 supplement (Gibco), 1× MEM non-essential amino acids solution (Gibco), 2 µg/ml DOX (Sigma-Aldrich, dissolved in water) and 10 µM ROCK inhibitor (Tocris) added on the day of seeding (Fernandopulle et al., 2018), with daily medium changes for two more days. On the fourth day, they were split using Accumax (Invitrogen) and replated onto plates coated with 0.1 mg/ml poly-D-lysine (Gibco) and 10 µg/ml laminin (Sigma-Aldrich). At this stage, I3Ns are at DIV3. 50,000 cells were seeded on 13 mm acid-etched glass coverslips (Lazo and Schiavo, 2023) for imaging, 150,000 cells were added to the somatic side of MFCs and 10,000,000 were used per 10 cm dish for proteomics. In the latter case, dishes were additionally coated with Geltrex to improve cell attachment. Neuronal maintenance medium consisted of Brainphys (StemCell Technologies) supplemented with 1× B27 (Gibco), 1× N2 (Gibco), 10 ng/ml recombinant brain-derived neurotrophic factor (PeproTech), 10 ng/ml recombinant glial cell line derived neurotrophic factor (PeproTech) and 1 µg/ml laminin. Additionally, 2 µg/ml DOX and 1× CultureOne supplement (Gibco) were added on the first day after replating, with half-medium changes every 3–4 days. For proteomics experiments, Geltrex (1 in a 100) was additionally added to the medium.

### Lentivirus production
Lenti-HEK293T cells were used to produce lentiviruses using VSV-G (Addgene #12259) and PAX (Addgene #22036) plasmids. Cells were cultured in Dulbecco's modified Eagle's medium (DMEM, Gibco) supplemented with 10% heat inactivated foetal bovine serum (FBS) and 1× GlutaMAX (Gibco). The medium was collected twice starting 2 days after transfection with Lipofectamine 3000 (Invitrogen) and concentrated using Lenti-X concentrator (Takara Bio), following manufacturers' instructions. Lentiviruses were titrated in I3Ns using increasing viral dilutions (1:100–1:2000) and tested by immunofluorescence to identify appropriate concentrations maximising transduction with minimal cell death.

### Microfluidic chambers
MFCs were prepared following published protocols (Restani et al., 2012; Panzi et al., 2023) and baked in resin moulds at 65°C for 1 h, sterilised with 70% ethanol and attached to plasma cleaned glass-bottom dishes (WillCo Wells), followed by further sterilisation by UV light for 10 min.

### Plasmids
All plasmids used in this study were generated using Gibson assembly (HiFi 2× Mastermix, NEB), following the manufacturer's guidelines. Inserts were PCR-amplified using Q5 polymerase master mix (NEB) and plasmid backbones were linearised by restriction digestion (Table S2). The linearised backbone was digested with DpnI (Invitrogen), and both backbone and insert DNA were purified using SPRI beads (Mag-Bind Total Pure NGS, Omega Biotech). Following assembly, constructs were purified using SPRI beads, transformed into DH5α competent cells (NEB) and verified by whole-plasmid nanopore sequencing. For lentiviral plasmids, verified constructs were transformed into NEBStable (NEB) or One Shot Stbl3 (Invitrogen).

### Immunofluorescence
For immunofluorescence (IF) experiments, cells were fixed in 4% paraformaldehyde (PFA), 4% sucrose in PBS for 15 min at room temperature (RT), followed by an incubation for 1 h at RT in blocking solution [4% bovine serum albumin (BSA) and 0.1% saponin in PBS]. This was followed by 2 h incubation at RT (or overnight at 4°C) with primary antibodies diluted in antibody buffer (4% BSA, 0.05% saponin in PBS), washed with PBS and incubated for 1–2 h with secondary antibodies conjugated to Alexa Fluor dyes (Invitrogen) diluted 1:1000 in the same buffer. Antibodies are listed in Table S3.

Imaging was carried out on LSM 980 confocal microscope (Zeiss) using an oil immersion Plan-Apochromat 63×/1.40 NA objective or an oil immersion EC Plan-Neofluar 40×/1.30 NA objective. Selected samples were imaged using Airyscan 2 and processed with 3D Airyscan deconvolution software using default settings on Zen Blue (Zeiss). Shifts in the z-plane were measured

Journal of Cell Science

using fluorescent microbeads with the same imaging conditions as the experiment and corrected prior to image analysis. For LOV-Turbo light-dependent activity, the images were acquired on an Olympus IX3 Series (IX83) inverted microscope using a Yokogawa W1 spinning disk equipped with a Hamamatsu Orca Fusion CMOS camera. For the co-transport and control experiments verifying the behaviour of LAMP1 fused with different tags, imaging was carried out using a Nikon SoRa spinning disk confocal, equipped with Photometrics Prime 95B for simultaneous two-colour imaging, and a Nikon 60× (1.42 NA) or 100× (1.45 NA) oil objectives. Subsequent image processing was done in Fiji ImageJ (Schindelin et al., 2012). Unless otherwise stated, all images presented are maximal projections of a z-stack.

### Proximity ligation assay

PLA was carried out following manufacturer's instructions using a Duolink proximity ligation assay (kit Sigma-Aldrich), using cells fixed in 4% PFA as above, and permeabilised for 5 min in methanol at −20°C. After the PLA protocol was completed, cells were incubated with DAPI in PBS for 10 min at RT. A LAMP1-only control was included with every LAMP1 experiment; however, one RAB7A PLA was lost during processing and needed to be repeated alongside the control, hence there are four replicates for the LAMP1-only control and three for every protein tested.

For analysis, we used a custom Fiji ImageJ pipeline to process all the images applying the following set of steps. All channels were first maximally projected and background subtraction was performed on the PLA channel using a rolling ball radius=2. A Gaussian filter (sigma=0.5/1) was applied to the background subtracted PLA channel, followed by auto-thresholding using Renyi's entropy method (Sahoo et al., 1997) and further refined using a watershed algorithm to separate any overlapping PLA puncta. Puncta were counted using the 'Analyze Particles' function, with an upper size limit of $0.2/0.3$ $\mu m^2$. For the DAPI channel, the contrast was enhanced to enable all nuclei to be counted. A Gaussian filter (sigma=2) was then applied to define the nuclei. Thresholding was performed using Huang's auto-thresholding followed by a watershed filter to separate touching nuclei. Nuclei were counted using the 'Analyze Particles' function, with a lower size limit of 45 $\mu m^2$. Images were then manually assessed, and nuclei counts were corrected where necessary. Images where the segmentation failed were removed from subsequent analysis. Data were exported as a .csv file for further processing in R (version 4.3.1) and RStudio (RStudio: Integrated Development Environment for R; Posit Software, PBC, Boston, MA, USA; https://posit.co/).

### Intensity profile blots

We used the line drawing feature in Fiji ImageJ to add a 10 $\mu m$ line (width=20 points) tracing a section of a neuron with colocalised puncta to make intensity profiles. The intensity profile plot tool was used to measure the intensity per point for the two channels of interest. Data were then analysed using R and RStudio. Intensity was scaled per channel using the following equation: normalised $x=x-minimum(x) / [maximum(x)-minimum(x)]$. The values were plotted using the geom_smooth function from Ggplot2 package (Wickham, 2016) with a span of 0.085. This fits a smoothed line to the plotted points using the locally estimated scatterplot smoothing (LOESS) function, where the span is twice the distance between consecutive intensity measurements.

### Axonal transport analysis

I3Ns cultured in MFCs were transduced with lentiviruses to express either LAMP1–EGFP or LAMP2A–mScarlet on DIV10 and transport was visualised on DIV13. Prior to imaging, HEPES-NAOH buffer, pH 7.4, was added to the medium at a final concentration of 20 mM. The imaging was carried out in an environmental chamber maintained at 37°C. 250 frames were captured at a frame rate of 2 frames/s, averaging four frames imaged as fast as possible in a 500 ms interval, with 0.085×0.085 $\mu m$ pixel size, zooming on axons in the distal part of the microgrooves where visible moving organelles could be detected. For co-transport experiments, I3Ns were co-transduced with lentiviruses encoding LAMP1–EGFP and LAMP2A–mScarlet. For the LAMP1 tag-control experiments, I3Ns in separate MFCs were transduced with either LAMP1–mScarlet or LAMP1–EGFP in one replicate, whereas in an independent replicate, axons from neurons expressing only LAMP1–EGFP were selected from co-transduced cultures for analysis. Directly prior to imaging, the media was changed to

phenol-free Brainphys medium (STEMCELL Technologies) prepared with the same concentrations of growth factors and supplements as previously described. Imaging was carried out in an environmental chamber maintained at 37°C with 5% $CO_2$, with up to 10 frames/s for 2 min and a 0.18×0.18 $\mu m$ pixel size.

Organelle speed and directionality were analysed using Trackmate (Tinevez et al., 2017; Ershov et al., 2022). All organelles that moved for at least five frames and across a third of the field of view were tracked, taking a maximum of 10 tracks following pausing, if organelles terminally paused. Stationary organelles that did not move at all during the recording were rare and were not tracked. Trackmate outputs three types of files: spots.csv, edges.csv and tracks.csv. The edges.csv file was used to calculate frame-to-frame directionality with negative consecutive displacements in the x-axis indicating retrograde transport, and positive indicating anterograde transport. This information was copied into the spots.csv file and was used to plot cumulative displacement of each organelle over time. Overall directionality of an organelle was inferred from the direction of the final displacement subtracted from the location in x-axis on the first frame, negative values were classified as retrograde and positive as anterograde. The mean speed of an organelle over the course of its movement was taken from the tracks.csv file. These operations were carried out using custom Python scripts, and subsequent statistical analysis and graphs were made using R and RStudio.

For co-transport analysis, kymographs were generated from 100 $\mu m$ axonal segments using KymographBuilder plugin in Fiji ImageJ, after background subtraction and Gaussian filtering (sigma=1) were applied to the images. At least 10 axons per replicate were selected, ensuring LAMP1–EGFP and LAMP2A–mScarlet were co-expressed. Axonal movements were manually traced, with tracks generated separately for each fluorophore. Tracks were colour coded by protein and overlaid onto a blank image in Fiji. Images corresponding to each channel were then combined using the Image Calculator function, with overlapping tracks appearing in black. Tracks were manually counted and scored as single-labelled or overlapping and classified for directionality. All subsequent statistical analyses and graphing were performed using R and RStudio.

### Statistical analysis

For the transport data, we fit a linear mixed-effects model including replicates as a random effect. We used a likelihood ratio test comparing two models, a full model including an interaction term between directionality and marker protein (LAMP1 or LAMP2A) to a reduced model without the interaction term, which showed that the full model was a significantly better fit for the data. Therefore, we included an interaction term between directionality and marker protein (LAMP1 or LAMP2A) in the linear mixed-effects model. We compared the velocity between LAMP1 and LAMP2A using an estimated marginal means (Emmeans) test, which is robust to outliers, utilising the fitted model to account for variability. The same test was used for the PLA analysis, with P-values adjusted for multiple testing using the false discovery rate (FDR) method of Benjamini and Hochberg. The package used for statistical analysis is Rstatix (rstatix: Pipe-Friendly Framework for Basic Statistical Tests; R package version 0.7.2; https://rpkgs.datanovia.com/rstatix/).

### Western blotting

Cells were lysed in 50 mM Tris-HCl pH 7.5, 100 mM NaCl, 1% Igepal CA-630, 0.1% SDS and 0.5% sodium deoxycholate containing 1× Halt protease inhibitor cocktail with EDTA (Thermo Fisher Scientific) for 15 min at 4°C. Lysates were then clarified by centrifugation at 21,800 $g$ at 4°C for 20 min. Clarified cell lysates were boiled for 10 min at 96°C in 1× NuPAGE LDS loading buffer (Invitrogen) with 100 mM dithiothreitol (DTT). Lysates were resolved by SDS-PAGE using 4-12% NuPAGE Bis-Tris Mini Protein Gels (Invitrogen) and nitrocellulose membranes (Whatman) using semi-dry transfer kit (Bio-Rad) or by wet transfer using the Novex wet transfer system (Thermo Fisher Scientific), following the manufacturers' instructions. Protein concentrations were measured using Bradford assay (Thermo Fisher Scientific) following the manufacturer's instructions. For streptavidin blots, beads were boiled in 1× NuPAGE LDS loading buffer to release biotinylated proteins. Membranes were blocked for 1 h in 5% skimmed milk in 0.1% Tween-20 in Tris-buffered saline (TBST) and incubated overnight

at 4°C with primary antibodies diluted in blocking buffer (Table S3). For Extravidin–HRP (horseradish peroxidase) or IRDye 680RD–Streptavidin, the blocking buffer was 5% BSA in TBST as milk contains biotin. After primary antibody incubation, membranes were washed three times in TBST followed by an incubation with a secondary HRP-conjugated antibody diluted 1:1000 in blocking buffer for 90 min. Blots were visualised using Bio-Rad Chemidoc Touch following 1 min incubation with Immobilon Classico Western HRP substrate (Millipore) or Odyssey CLx LI-COR 4.11 for blots incubated with IRDye. Uncropped blots from this work are shown in Fig. S7.

### Blue light box

The blue light box was made by laser-cutting a 5 mm thick black acrylic sheet to assemble a 25 cm×25 cm×10 cm box that can fit four 10 cm dishes side-by-side. The inside of the box contains a 3 mm thick frosted acrylic sheet cut slightly smaller to fit into the box. For the circuit 16 super-bright LEDs (∼2.5 V measured forward voltage, 50 lumens) were connected in parallel to an Arduino Nano with a ∼11 Ω resistor in series with the LEDs and the Arduino Nano. A transistor (TIP120) was added to allow more current to reach the LEDs without compromising the Arduino Nano and a voltage regulator (Wurth Elektronik, 7806CT) was added to ensure the voltage drop across the LEDs remains consistent. The frosted acrylic sheet rests on thin styrofoam supports glued at the corners of the box. The circuit with the Arduino Nano was glued to the side of the box to minimise interference with light. Using a Thorlab PM100D with a S120VC sensor, we measured a light power of ∼135 $\mu$W/cm$^2$.

### Proximity labelling

Biotin (Sigma-Aldrich) was made as a 250 mM stock in DMSO and stored in aliquots at −20°C until use. In all biotinylation experiments in I3Ns, the cells were transduced on DIV15 with LOV-Turbo lentiviruses at an appropriate dilution, and when DOX-inducible NES-LOV was used, the medium was supplemented with 200 ng/ml DOX. On DIV18, biotin was added to I3Ns at a final concentration of 250 $\mu$M and the cells were immediately exposed to blue light pulses for 30 min, followed by three washes with cold PBS before snap-freezing. For experiments using DOX-inducible NES-LOV in Lenti-HEK293T cells, cells were transfected with 500 ng of the plasmid using Lipofectamine 3000, and the medium supplemented with 200 ng/ml DOX for 24 h before exposure to blue light and cell lysis.

For MS experiments, I3Ns were washed by cold PBS and pelleted by centrifugation at 1000 $g$ for 1 min at 4°C and snap-frozen in liquid nitrogen. Prior to pulldown of biotinylated proteins, pellets were lysed in 1 ml cold lysis buffer with 1× cOmplete protease inhibitor cocktail (Roche) followed by a 45 min incubation shaking at 800 rpm in a cold room set at 4°C. Lysates were then cleared by centrifugation at 21,800 $g$ for 20 min at 4°C and supernatants collected. For pulldown of biotinylated proteins, Pierce magnetic A/G beads (Thermo Fisher Scientific) were used to pre-clear the lysates and Pierce magnetic streptavidin beads (Thermo Fisher Scientific) to pulldown biotinylated proteins. Beads were washed three times in cold lysis buffer prior use for pulldowns/pre-clearing in protein LoBind tubes (Eppendorf). For the pre-clearing step, equal amounts of proteins per condition were added to 50 $\mu$l A/G beads and incubated on a rotating wheel for 45 min at 4°C. The supernatants were then incubated with 100 $\mu$l streptavidin beads on a rotating wheel overnight at 4°C, keeping ∼25 $\mu$l of the pre-cleared lysates for a western blot. The beads were washed three times with cold lysis buffer, three times with high salt buffer (50 mM Tris-HCl pH 7.5, 1 M NaCl, 1 mM EDTA, 1% Igepal CA-630, 0.1% SDS and 0.5% sodium deoxycholate) and once with 50 mM cold NH$_4$HCO$_3$ pH 8. The beads were then transferred into a new tube and washed again twice with NH$_4$HCO$_3$.

### MS sample preparation and acquisition

Bead-bound proteins were prepared for mass spectrometric analysis by in solution enzymatic digestion. Briefly, bead-bound proteins in 50 mM NH$_4$HCO$_3$ were reduced in 10 mM DTT and then alkylated with 55 mM iodoacetamide. After alkylation, 0.4 $\mu$g of trypsin (Thermo Fisher Scientific) was added and the proteins digested overnight at 37°C in a thermomixer (Eppendorf, Germany), shaking at 800 rpm. After digestion,

1 $\mu$l of formic acid (FA) was added and the beads centrifuged for 60 s at 2000 $g$, before placing them in a magnetic rack. The supernatant was transferred into a fresh tube and desalted off-line using C18 trap columns (EV2018 Evotip Pure, Evosep Biosystems, Denmark). Peptides were eluted with 50% acetonitrile (ACN), vacuum-dried and resuspended in 30 $\mu$l 0.1% (v/v) FA prior to LC-MS/MS analysis.

Peptides were analysed by nano-scale capillary LC-MS/MS using an Ultimate U3000 HPLC (ThermoScientific Dionex, San Jose, USA) to deliver a flow of ∼300 nl/min. A C18 Acclaim PepMap100 5 $\mu$m, 75 $\mu$m×20 mm nanoViper (Thermo Fisher Scientific), trapped the peptides prior to separation on an EASY-Spray PepMap RSLC 2 $\mu$m, 100 Å, 75 $\mu$m×500 mm nanoViper column (Thermo Fisher Scientific). Peptides were eluted at a constant flow rate of 0.300 $\mu$l/min using the following 90 min gradient: 2% Buffer B (75% ACN, 5% DMSO, 0.1% FA in water) for 0–6 min, 8–55% Buffer B from 7–67 min, 95% Buffer B from 67.5–74 min and re-equilibrated at 2% Buffer B from 75–90 min. Buffer A was 5% DMSO and 95% 0.1% FA in water. The analytical column outlet was directly interfaced via a nano-flow electrospray ionisation source, with a hybrid dual pressure linear ion trap mass spectrometer (Orbitrap Lumos, Thermo Fisher Scientific). Data-dependent analysis was carried out, using a resolution of 120,000 for the full MS spectrum, followed by as many subsequent MS2 scans on selected precursors as possible within a 3 s maximum cycle time. MS1 was performed in the Orbitrap instrument with an AGC target of 4×10$^5$, a maximum injection time of 50 ms, and a scan range from 375 to 1500 $m/z$. MS2 was performed in the ion trap with a rapid scan rate, an AGC target of 2×10$^3$, and a maximum injection time of 300 ms. Isolation window was set at 1.2 $m/z$, and 32% normalised collision energy was used for HCD. Dynamic exclusion was used with a time window of 40 s.

### MS data analysis

All raw files were then analysed using Fragpipe (Yu et al., 2021; Yang et al., 2023) set at the default 'Label-free quantification' workflow using MSFragger (Kong et al., 2017) against the reviewed UniProt human proteome (UP000005640). LOV-Turbo (translated from amino acid sequence) and streptavidin (P22629) were manually added. MSFragger options were used at default settings with trypsin as the digestion enzyme allowing up to two missed cleavages. Cysteine carbamidomethylation was set as a fixed modification, whereas oxidation of methionine, acetylation of N-termini and biotinylation of lysine (Unimod accession #3) were set as variable modifications. FDR was set at 1% at the peptide level. Reverse decoys were generated for all sequences by Fragpipe and used to calculate the FDR. MSbooster, IonQuant and match-between-runs (MBR) were all used with MBR min correlation=0 and MBR toprun=10, and FDR set at 1%.

The MSstats file generated was used in RStudio for subsequent analysis using MSstats (version 4.8.7) (Kohler et al., 2023), with normalisation using 'globalStandards' (LOV-Turbo, PCCA, PCCB and PYC) and summarisation using all features with Tukey's median polish. Data were imputed using MBimpute. For quantitative analysis of differential protein abundance, a comparison matrix was generated for Batch 1 and Batch 2 MS runs using the MSstats groupComparison function, applying 0.5 weights for batch 1 and batch 2, with pairwise comparisons between all conditions. For comparison, the degree of protein enrichment in the UT condition over all other conditions (LAMP1, LAMP2A or NES) was calculated. If a protein was found enriched in the UT condition in one or more comparisons (log$_2$ fold-change>1.5 and adjusted $P$<0.05), it was filtered out of the full dataset. Remaining proteins were taken forward for further analysis, comparing LAMP1 and LAMP2A respectively to the cytoplasmic NES control. Any proteins that had a fold-change>1.5 and adjusted $P$-value<0.05 were selected for another screen, selecting from this list proteins that were also enriched over the UT control. Volcano plots were generated using Ggplot, with test_gsea and plot_PCA code from DEP package (Zhang et al., 2018) to carry out and plot GSEA and PCA. For Venn diagrams, the VennDiagram package (Chen and Boutros, 2011) was used.

### Artificial intelligence use

ChatGPT (OpenAI) was used to assist in generating analysis code presented in this manuscript. It was also used in a limited capacity for manuscript

editing, such as rewording and summarising text to improve readability. After using these services, the authors reviewed and edited the content as needed and take full responsibility for the content of the publication.

### Acknowledgements
We are grateful to Dr James Sleigh and Dr David Villarroel Campos (University College London), and Dr Flora Lee (King's College London) for helpful discussions and feedback. We thank Dr Gavin Kelly (Bioinformatics & Biostatistics STP, The Francis Crick Institute) for advice on statistical analysis, and Dr Nic Cade (UKDRI) for guidance with imaging. All microscopy was carried out at the UK Dementia Research Institute Imaging facility (UCL) and the Advanced Light Microscopy STP (The Francis Crick Institute). Proteomics experiments were supported by the Proteomics STP (The Francis Crick Institute). We also thank members of the Ule and Schiavo laboratories for their valuable input and support over the years.

### Competing interests
The authors declare no competing or financial interests.

### Author contributions
Conceptualization: R.A., J.U., G.S.; Data curation: R.A., A.M.A., F.I.; Formal analysis: R.A.; Funding acquisition: R.A., J.U., G.S.; Investigation: R.A.; Methodology: R.A., O.G.W., S.-Y.L., M.S., A.T.; Software: R.A.; Supervision: N.B., J.U., G.S.; Validation: R.A.; Visualization: R.A., G.S.; Writing – original draft: R.A., G.S.; Writing – review & editing: R.A., A.M.A., O.G.W., S.-Y.L., F.I., M.S., A.T., N.B., J.U., G.S.

### Funding
This work was supported by the Wellcome Trust (4-year PhD studentship 175261 to R.A., Investigator Award 223022/Z/21/Z to G.S. and Joint Investigator Award 215593 to J.U.), the UK Dementia Research Institute (UKDRI-1005 to G.S., and UKDRI-RE21605 to J.U.), the Motor Neurone Disease Association (186479 to A.M.A., and Birsa/Oct21/976-799 to N.B.) and the Lady Edith Wolfson Fellowship funded by the Motor Neuron Disease Association and the Rosetrees Charity to O.G.W., the GPCR collaborative of the St. Jude Children's Research Hospital to A.T., the Daegu Gyeongbuk Institute of Science and Technology (DGIST) Start-up Fund Program of the Ministry of Science (ICT 2025020055) to S.-Y.L., the Korean Ministry of Trade, Industry and Energy (MOTIE) and the Korea Institute for Advancement of Technology (KIAT) through the 'International Cooperative R&D Program' (Task No. P0028350) to G.S. This work was supported by the Francis Crick Institute, which receives its core funding from Cancer Research UK (CC0102, CC1063), the UK Medical Research Council (CC0102, CC1063), and the Wellcome Trust (CC0102, CC1063) [F.I. and M.S.]. Open Access funding provided by University College London. Deposited in PMC for immediate release.

### Data and resource availability
MS data have been deposited to the ProteomeXchange Consortium via the PRIDE (Perez-Riverol et al., 2025) partner repository with the dataset identifier PXD067562. All scripts used to generate figures in this manuscript have been deposited on a repository on Github (https://github.com/reem-abouward/LAMP-manuscript.git).

All other relevant data and details of resources can be found within the article and its supplementary information.

### First Person
This article has an associated First Person interview with the first author of the paper.

### Peer review history
The peer review history is available online at https://journals.biologists.com/jcs/lookup/doi/10.1242/jcs.264466.reviewer-comments.pdf

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
