## [Peer Review File · Journal of Cell Science]

LAMP1 and LAMP2A localise to axonal organelles with distinct motility dynamics and partially overlapping molecular signatures in human neurons

Reem Abouward, Alya Masoud Abdelhafid, Oscar G. Wilkins, Song-Yi Lee, Fairouz Ibrahim, Mark Skehel, Alice Ting, Nicol Birsa, Jernej Ule and Giampietro Schiavo
DOI: 10.1242/jcs.264466

Editor: Subhojit Roy

Review timeline

Original submission:	19 September 2025
Editorial decision:	23 October 2025
First revision received:	6 January 2026
Accepted:	8 January 2026

Original submission

First decision letter

MS ID#: jcs.264466

MS TITLE: Characterisation of LAMP1- and LAMP2A-positive organelles in neurons

AUTHORS: Reem Abouward; Alya Masoud Abdelhafid; Oscar G Wilkins; Song-Yi Lee; Fairouz Ibrahim; Mark Skehel; Alice Ting; Nicol Birsa; Jernej Ule; Giampietro Schiavo
ARTICLE TYPE: Research Article

Dear Dr Schiavo,

Thank you for submitting, it is a very nice paper and JCS would be happy to publish it after revisions. I share Rev 1's sentiment that the paper would have more impact if it was written in a way that highlighted the findings better - encapsulated in Rev #1's comment "...the different experimental parts were only linked thematically as an observational study rather than a deterministic test of a central hypothesis". For example, even the title is too generic and does not really convey what was actually discovered. Please consider spending some time to rewrite parts of the paper using more direct language - avoiding somewhat wishy-washy statements - because I strongly think that this will increase the visibility of this important work. Please provide a point by point rebuttal by addressing all the reasonable issues raised by the reviewers, or provide a strong rationale for items that you may feel beyond scope or best addressed in future studies.

Reviewer 1

This study by Abouward et al is a characterisation in hiPSC neurons of LAMP1 and LAMP2A vesicles. They show that these two markers are not necessarily interchangeable and they analyse the movement of the organelles as well as their interactomes. The work is high quality and the datasets are likely to be useful for neuronal cell biologists.

It did feel to me that the paper lacks a punchline. In the abstract it states that the interactome work is undertaken to "To explore the molecular mechanism underlying [the differences in movement]" but there isn't any insight into these differences that emerges from the lists

uncovered. The work is valuable in itself, but the different experimental parts were only linked thematically as an observational study rather than a deterministic test of a central hypothesis. The text could be edited to better reflect that.

1. The main idea is that LAMP2A-positive organelles are a subset of LAMP1-positive organelles. This is mainly because the proteomics detects a subset of LAMP1 interactors in the LAMP2A dataset. Isn't this more likely to be due to coverage (there are less interactors detected overall)? Rather than the LAMP2A-positive vesicles being a bona fide subset of LAMP1-positive vesicles?
2. Related to this, the initial imaging is really nice but this experimental setup could easily be used to test the idea about subsets of LAMP-positive vesicles. Co-imaging of LAMP1-GFP and LAMP2A-mScarlet would tell us about cotraffic and subsets of vesicles. Have the authors attempted this? As a side note the Figure says that the proteins are co-transduced, whereas the text and legend say that transduction is with either LAMP1-EGFP or LAMP2A.
3. Again, on this point, do the authors know whether the tag itself affects the speed analysis? Is the retrograde motion of LAMP1-mScarlet also slower than LAMP2A. Is LAMP2-GFP faster than LAMP1?
4. I'm not a fan of using line profiles to look at colocalisation. Since it is very much dependent on where the line is drawn. In Figure 4, we see lots of green and magenta and very little white. Which suggests that the majority of SYT1 or SNAP25 is not colocalised with either LAMP1 or LAMP2. We do not expect high levels of colocalisation but even in the examples shown, such as in A, the green spot is close to the LAMP1 spot but doesn't colocalise with it. In C the straightened version is shown with different brightness and contrast such that the white signal (overlay) disappears. On the whole I didn't find this figure very convincing. Can the authors do a larger scale analysis on their data? It is possible to use line profiles (so that the correct neurites are analysed) but segment them such that peaks are isolated and then the overlap of regions can be calculated.

For the interactome data (Fig 3), the authors could try to plot the enrichment in LAMP1 vs LAMP2A for the 112 overlapping proteins. It would be interesting to see if this abundances correlate (test with Pearson's). It looks like SNX3 and VAMP3 are high abundance in both but there may be some differences between the two datasets that could be important?

Reviewer 2

In their manuscript the authors have used an in vitro i3N human neuronal culture to characterize Lamp1 vs Lamp2a-containing organelles with respect to transport dynamics as well as their local interactomes. For this they used proximity labeling based on LOV-Turbo which requires blue light activation to biotinylate nearby proteins. This resulted in Lamp1- and Lamp2a-specific interactomes, that however largely overlapped. The overall conclusion is that Lamp2a-lysosomes may represent a subset of the total pool of Lamp1 positive organelles, and this was validated for a set of common hits for both pools. The technical and experimental quality is high, but there are a number of concerns that need to be addressed and may require additional experiments or scrutiny.

SUGGESTIONS TO AUTHORS

- this Reviewer is missing a rationale and clear link between the trafficking analysis focused on axons and the interactome analyses, performed in whole neurons. As the authors indicate the distinct Lamp1 and Lamp2a positive organelles may have different roles in dendrites and axons, and this distinction is not made anymore in the experimental approach. It would have been more logic, also in the relevance for neurodegenerative diseases, to have kept the focus on axons, using microfluidics. Secondly, the axonal transport shows very dynamic Lamp1/Lamp2a organelles, with hardly stationary ones. Could this be due to the fact that the authors look at relatively young neurons (which is also problematic to extrapolate this to NDs): whereas for primary hippocampal neurons DIV14 and later are mostly used, in the case of iNeurons, differentiation to mature neurons (including active synapses) takes overall longer, and it is advisable to perform experiments at DIV25 and more. The authors cannot exclude the possibility that more distinct pools are present in older, mature neurons. Related to this, when overlooking the images with biotinylated proteins, the vast majority of biotin labeling (eg fig 2) localizes to the soma, as shown by very strong labeling compared to V5 and Lamp1 or Lamp2a: from my experience the pool of lysosomes in the perinuclear region is rather stationary, and the interactome analyses may be biased towards this stationary pool and therefore do not ideally allow to differentiate between Lamp1 and Lamp2a

lysosomes in neurites (and which have been shown to have distinct functions). Focusing on the axonal interactomes would overcome this bias.

- The authors apply LOV-Turbo, arguing the toxicity associated with H₂O₂. However, LOV-Turbo requires 30min of blue light labeling, and pulsed light is used to minimize overheating; hence not avoiding it. In Turbo-ID, the 1min of H₂O₂ may cause less damage and at least would not compromise temporal changes that may occur during 30 min of labeling (caused by overheating). Also, a 30min labeling not only abolishes temporal resolution, but would result in stripes of biotinylation as individual organelles move a significant distance over this time period. This is not evident from the figures provided in Fig.2 and the rather heavy granular labeling in the soma further underscores the larger pool of stationary lysosomes (see comment above). Related to Fig2E-F, one would expect as well an identical staining for V5 and Lamp1/Lamp2a which is not the case for both sets of data. Particularly for the Lamp2a there is essentially no overlap with V5 and in this case the biotin label also doesn't match Lamp2a (and several strong spots for Lamp1 are not positive for biotin, nor V5). The authors should provide more convincing images that the model and experimental approach is reliably capturing biotinylation on lysosomes.

- With respect to the interactome analyses, it should be more clearly mentioned in the text that experiments were done in biological quadruplicate (only mentioned in the legend to figure 3). The samples were run twice (batch 1 and 2): is there a logic explanation why the PCA between the two batches of the same conditions differ that much? More importantly, with respect to validation, the authors focus on common interactors for both Lamp1 and Lamp2a. Given the focus is on different lysosomal populations, I would argue that it is more relevant to validate unique interactors for each population. Why is this not done? Can the authors identify unique proteins and validate these likewise. Are there interactors known to be uniquely localized to either axons or dendrites: scrutinizing this more in validation experiments would improve the impact of the manuscript. Secondly, among the common interactors, they validate synaptic proteins, including Syt1 and Snap25. However, the imaging data only show Syt1 and Snap25 positive spots of which some are closely associated with Lamp1 and Lamp2a. At this resolution, one would expect to see clearly overlapping spots if they are co-localized on the same organelle. The data are not convincing to state that they represent biosynthetic Lamp organelles carrying synaptic proteins; the authors should include live imaging to demonstrate they co-traffic as well. Moreover, the authors should perform validation experiments in their microfluidic chamber to provide more conclusive data on axonal colocalization and co-trafficking. Of note, the authors conclude in their abstract that LAMP2a are a subpopulation of the Lamp1 organelles; but in their Discussion they also state that Lamp1 and Lamp2a label distinct organelle populations. From the interactome data one could favor the former, whereas the transport kinetics suggest the latter. They only found 8 proteins unique for Lamp2a, among which is Lamp2: does that not demonstrate it is a distinct pool? If part of Lamp1 population, and given Lamp proteins are abundant, Lamp2 should have been recovered in the Lamp1 interactome, no? Can the authors clarify the discrepancy in their conclusions? An easy experiment could be to perform co-immunostaining for both lamps.

First revision

Author response to reviewers' comments

Dear Dr Schiavo,

Thank you for submitting, it is a very nice paper and JCS would be happy to publish it after revisions. I share Rev 1's sentiment that the paper would have more impact if it was written in a way that highlighted the findings better - encapsulated in Rev #1's comment "...the different experimental parts were only linked thematically as an observational study rather than a deterministic test of a central hypothesis". For example, even the title is too generic and does not really convey what was actually discovered.

Please consider spending some time to rewrite parts of the paper using more direct language - avoiding somewhat wishy-washy statements - because I strongly think that this will increase the visibility of this important work.

We thank the Editor for the kind words and the encouragement to amend the manuscript implementing a more direct narrative. We therefore have decided to change the title to *“LAMP1 and LAMP2A localise to axonal organelles with distinct motility dynamics and partially overlapping molecular signatures in human neurons”*.

Please provide a point by point rebuttal by addressing all the reasonable issues raised by the reviewers, or provide a strong rationale for items that you may feel beyond scope or best addressed in future studies.

Reviewers' Comments

Reviewer 1

This study by Abouward et al is a characterisation in hiPSC neurons of LAMP1 and LAMP2A vesicles. They show that these two markers are not necessarily interchangeable and they analyse the movement of the organelles as well as their interactomes. The work is high quality and the datasets are likely to be useful for neuronal cell biologists.

We thank the Reviewer for the kind words and appreciation of our study.

It did feel to me that the paper lacks a punchline. In the abstract it states that the interactome work is undertaken to "To explore the molecular mechanism underlying [the differences in movement]" but there isn't any insight into these differences that emerges from the lists uncovered. The work is valuable in itself, but the different experimental parts were only linked thematically as an observational study rather than a deterministic test of a central hypothesis. The text could be edited to better reflect that.

We thank the Reviewer for the great suggestion. We have reworded the text to be more speculative and have increased the punchline. We changed the title to *“LAMP1 and LAMP2A localise to axonal organelles with distinct motility dynamics and partially overlapping molecular signatures in human neurons”* and reworded the abstract to stress the novelty and implications of our study.

1. The main idea is that LAMP2A-positive organelles are a subset of LAMP1-positive organelles. This is mainly because the proteomics detects a subset of LAMP1 interactors in the LAMP2A dataset. Isn't this more likely to be due to coverage (there are less interactors detected overall)? Rather than the LAMP2A-positive vesicles being a bona fide subset of LAMP1-positive vesicles?

We agree with the Reviewer that the apparent subset relationship observed in the proteomic datasets is likely to be influenced by coverage limitations. The LAMP2A proximity labelling experiments detect fewer interactors overall, which could bias the overlap toward a subset of the LAMP1 interactome rather than demonstrating that LAMP2A-positive vesicles constitute a bona fide subset of LAMP1-positive organelles. Indeed, our axonal transport experiment suggest overlapping but distinct pools for both LAMP1- and LAMP2A-positive organelles, which are characterised by different motility behaviours.

Following the Reviewers' suggestions, we carried out transport experiments by co-transducing neurons with both LAMP1 and LAMP2A, which revealed incomplete co-localisation during axonal transport with ~60% of motile organelles being positive for both proteins (double-carrier), and a higher degree of co-transport in the retrograde direction.

The proteomics experiment failed to capture these differences as we used as starting material neurons in mass culture. Thus, small (but meaningful) differences in the axons are likely to be obscured by cellular heterogeneity, saturated by somatic signal and noise. We attempted to combine LOV-Turbo proximity labelling with axonal enrichment; however, this approach proved technically challenging and did not yield robust axon-enriched interactomes. Other members of the laboratory have tried similar axonal proteomics and experienced similar difficulties.

In light of these considerations, we have revised the text to clarify these points, emphasising that whilst the proteomic data suggest that LAMP2A-positive organelles are a subset of LAMP1 organelles, the transport data reveal a different picture, indicating functionally distinct and partially overlapping organelle populations. These differences are not fully resolved by bulk proteomic analyses and require further method optimisations to enrich for axonal interactomics.

2. Related to this, the initial imaging is really nice but this experimental setup could easily be used to test the idea about subsets of LAMP-positive vesicles. Co-imaging of LAMP1-GFP and LAMP2A-mScarlet would tell us about cotraffic and subsets of vesicles. Have the authors attempted this? As a side note the Figure says that the proteins are co-transduced, whereas the text and legend say that transduction is with either LAMP1-EGFP or LAMP2A.

We thank the Reviewer for this suggestion and agree that LAMP1 and LAMP2A co-imaging would provide a direct test of co-transport. Indeed, we attempted such experiments previously; however, our earlier reliance on laser-scanning confocal microscopy limited our ability to image both fluorophores simultaneously, and to reliably quantify co-transport.

To overcome this limitation, we used for these experiments spinning-disk confocal microscopy, which allows imaging over a larger field of view with lower laser power and reduced phototoxicity and bleed-through. Using this setup, we performed co-transport experiments in neurons co-transduced with LAMP1-EGFP and LAMP2A-mScarlet, and quantified co-localised motile tracks in axons. This data is now included as Figure 1F and 1G, demonstrating partial co-transport of LAMP1- and LAMP2A-positive organelles, with greater co-transport observed in the retrograde direction.

We have also corrected the inconsistency noted by the Reviewer between the figure, text, and legend to clearly indicate when constructs were expressed individually versus co-transduced.

3. Again, on this point, do the authors know whether the tag itself affects the speed analysis? Is the retrograde motion of LAMP1-mScarlet also slower than LAMP2A. Is LAMP2-GFP faster than LAMP1?

EGFP and mScarlet are of comparable size and, in both cases, were fused to a short cytoplasmic region of LAMPs, making differential interference with protein function unlikely. Consistent with this, previous studies using different fluorescent tags have reported similar direction trends, with LAMP2A-positive organelles moving more slowly in the retrograde direction compared to anterograde transport; comparable trends were reported for LAMP1 in other systems.

Nonetheless, to directly address this concern in our system, we perform transport experiments directly comparing LAMP1-EGFP and LAMP1-mScarlet transduced in separate experiments. We observed no difference in retrograde transport speed between the two constructs, and only a modest difference in mean anterograde speed, which was associated with high variance (\pm -1 μ m/s). We expect this difference to significantly reduce with increased sampling.

Taken together, both our experimental data and prior literature indicate that the observed differences in transport dynamics between LAMP1- and LAMP2A-positive organelles reflect intrinsic properties of these proteins rather than the influence of the fluorescent tags.

4. I'm not a fan of using line profiles to look at colocalisation. Since it is very much dependent on where the line is drawn. In Figure 4, we see lots of green and magenta and very little white. Which suggests that the majority of SYT1 or SNAP25 is not colocalised with either LAMP1 or LAMP2. We do not expect high levels of colocalisation but even in the examples shown, such as in A, the green spot is close to the LAMP1 spot but doesn't colocalise with it. In C the straightened version is shown with different brightness and contrast such that the white signal (overlay) disappears. On the whole I didn't find this figure very convincing. Can the authors do a larger scale analysis on their data? It is possible to use line profiles (so that the correct neurites are analysed) but segment them such that

peaks are isolated and then the overlap of regions can be calculated.

As the Reviewer noted, most synaptic proteins such as SYT1 and SNAP25 do not colocalise with LAMP-positive organelles. This is expected given the high abundance of synaptic vesicle proteins and the much smaller size of synaptic vesicles (SVs)/SV precursor compartments relative to LAMP-positive organelles. When colocalisation is observed, synaptic proteins often appear to partially decorate the surface of LAMP-positive organelles rather than displaying a full overlap. These limitations prompted us to validate our interactome data using proximity ligation assays (PLA) with proteins that are less abundant and more spatially restricted, allowing for a more confident validation of proximity relationships identified in the proteomic datasets. Given these limitations, we have moved the original Figure 4 to Supplementary Information and revised the text to clarify the figure is intended to illustrate spatial proximity rather than extensive colocalisation.

For the interactome data (Fig 3), the authors could try to plot the enrichment in LAMP1 vs LAMP2A for the 112 overlapping proteins. It would be interesting to see if this abundances correlate (test with Pearson's). It looks like SNX3 and VAMP3 are high abundance in both but there may be some differences between the two datasets that could be important?

We appreciate this suggestion and agree that correlation analysis could, in principle, provide insight into differential enrichment between the LAMP1 and LAMP2A interactomes. However, in our datasets, such an analysis is technically limited by differences in detection depth and dynamic range between the two proximity labelling experiments. The LAMP2A dataset captured fewer interactors overall and exhibited lower fold-change values, with lower expression levels for LAMP2A-LOV Turbo. As a result, only a subset of shared proteins can be meaningfully assessed, and direct comparison of enrichment values is confounded by coverage limitations, making it difficult to distinguish true preferential association from differences in protein abundance or detection sensitivity.

Additionally, because the proximity labelling experiments were performed in mass cultures, axon-specific differences are likely obscured by much more abundant somatic signals. We therefore chose not to include this analysis in the current manuscript. We agree that this approach would be highly informative using more sensitive approaches, such as POTAToMap.

Reviewer 2

In their manuscript the authors have used an in vitro i3N human neuronal culture to characterize Lamp1 vs Lamp2a-containing organelles with respect to transport dynamics as well as their local interactomes. For this they used proximity labeling based on LOV-Turbo which requires blue light activation to biotinylate nearby proteins. This resulted in Lamp1- and Lamp2a-specific interactomes, that however largely overlapped. The overall conclusion is that Lamp2a-lysosomes may represent a subset of the total pool of Lamp1 positive organelles, and this was validated for a set of common hits for both pools. The technical and experimental quality is high, but there are a number of concerns that need to be addressed and may require additional experiments or scrutiny.

This Reviewer is missing a rationale and clear link between the trafficking analysis focused on axons and the interactome analyses, performed in whole neurons. As the authors indicate the distinct Lamp1 and Lamp2a positive organelles may have different roles in dendrites and axons, and this distinction is not made anymore in the experimental approach. It would have been more logic, also in the relevance for neurodegenerative diseases, to have kept the focus on axons, using microfluidics.

We agree with this Reviewer that, in principle, axon-restricted interactome analyses would provide a stronger and more direct link to the axonal trafficking data. We attempted to perform axon-enriched proximity labelling using the SOL3D platform (Hagemann *et al.*, 2024), which allows the harvest of large amounts of axonal material as opposed to

standard microfluidic devices. However, the enrichment of axonal proteins compounded with the pulldown of biotinylated material, was technically challenging and the resulting datasets were highly variable. Further optimisation of this approach is thus required and will be the focus of future studies.

To establish whether LAMP1- and LAMP2A-positive organelles share overlapping molecular identities, we employed proximity labelling in mass cultures, which we initially considered sufficient to capture core differences and similarities between their interactomes. Importantly, we complemented this experiment with axon-specific live-imaging analyses. This approach revealed that LAMP1- and LAMP2A-positive organelles form overlapping but distinct populations in the axons. Unfortunately, these differences are not readily resolved by proximity labelling performed in mass cultures. These findings highlight the limitations of bulk interactomic approaches and underscore the need for future studies using axon-enriched proteomic strategies for studying the identities of endo- lysosomal organelles.

Secondly, the axonal transport shows very dynamic Lamp1/Lamp2a organelles, with hardly stationary ones. Could this be due to the fact that the authors look at relatively young neurons (which is also problematic to extrapolate this to NDs): whereas for primary hippocampal neurons DIV14 and later are mostly used, in the case of iNeurons, differentiation to mature neurons (including active synapses) takes overall longer, and it is advisable to perform experiments at DIV25 and more. The authors cannot exclude the possibility that more distinct pools are present in older, mature neurons.

We thank the Reviewer for raising this important point. We do not conclude that stationary or pausing organelles are absent in our system, and indeed we attribute several of the observed differences between LAMP1- and LAMP2A-positive organelles to differences in pausing behaviour between the two populations. In addition, many organelles underwent “terminal” pausing following periods of processive movement. Distinguishing whether such events should be classified as paused or processive is inherently ambiguous; therefore, we included a consistent number of pausing frames in our analyses so that terminal pausing contributed to the average track velocity, while avoiding bias toward slower velocities that would arise from extended pausing.

Importantly, organelles not moving for the entire recording were rarely observed. We were therefore cautious in how such events were treated in our analyses. In our initial laser-scanning imaging setup, the field of view was limited, and recordings were relatively short (2 min), making it difficult to distinguish biologically meaningful terminal pausing from transient or laser-induced stochastic arrest. Properly resolving stationary behaviour would require substantially longer recordings at higher temporal resolution. Additionally, it is unclear in the field what constitutes pausing, with many publications defining it at arbitrary thresholds of 0.1 or 0.2 $\mu\text{m}/\text{s}$. Many of the “stationary” or “terminally paused” organelles we observed underwent stochastic sideways motion, which may be physiologically relevant. As this was outside the scope of this study, stationary organelles and detailed analysis of pausing dynamics were not the focus of our original analyses. Using the spinning-disk confocal setup, which enabled imaging over a larger field of view and at higher temporal resolution, we observed more frequent pausing events. In particular, at higher frame rates (10 fps), LAMP1-positive organelles exhibited a previously underappreciated propensity for brief, repetitive pausing events occurring on sub-second timescales. While this behaviour is potentially biologically interesting, we did not pursue its systematic quantification, as this would require dedicated analyses beyond the scope of the current study.

With respect to neuronal maturity, we acknowledge that human iNeurons are less mature than mouse primary hippocampal neurons, and that full synaptic maturation in iNeuron systems typically occurs at later time points (\geq DIV25). Nonetheless, available evidence suggests that axonal transport dynamics of several types of organelles are largely comparable between iNeurons and primary neurons. In particular, Boecker *et al.* (2020) reported broadly similar transport behaviours for endosomes, lysosomes, and other organelles in hippocampal neurons and iNeurons, with mitochondrial dynamics being the

only one showing changes with age in culture.

Based on these considerations, we selected DIV13 as a balance between neuronal maturation and experimental feasibility, and iNeurons were an excellent model to study human-specific differences. While we cannot exclude the possibility that more distinct organelle pools or altered pausing behaviours emerge at later developmental stages, the aim of the present study was to determine whether differences between LAMP1- and LAMP2A-positive organelles are evident at baseline. A systematic investigation of age-dependent changes, ideally coupled with measurements of synaptic connectivity and network activity, would represent an important and interesting direction for future work.

Related to this, when overlooking the images with biotinylated proteins, the vast majority of biotin labeling (eg fig 2) localizes to the soma, as shown by very strong labeling compared to V5 and Lamp1 or Lamp2a: from my experience the pool of lysosomes in the perinuclear region is rather stationary, and the interactome analyses may be biased towards this stationary pool and therefore do not ideally allow to differentiate between Lamp1 and Lamp2a lysosomes in neurites (and which have been shown to have distinct functions). Focusing on the axonal interactomes would overcome this bias.

We thank the Reviewer for raising this point and we agree that this is possibly why we were unable detect a unique LAMP2A pool. However, as discussed above, it was unfeasible to do the experiment on isolated axons.

- The authors apply LOV-Turbo, arguing the toxicity associated with H₂O₂. However, LOV-Turbo requires 30min of blue light labeling, and pulsed light is used to minimize overheating; hence not avoiding it. In Turbo-ID, the 1min of H₂O₂ may cause less damage and at least would not compromise temporal changes that may occur during 30 min of labeling (caused by overheating). Also, a 30min labeling not only abolishes temporal resolution, but would result in stripes of biotinylation as individual organelles move a significant distance over this time period. This is not evident from the figures provided in Fig.2 and the rather heavy granular labeling in the soma further underscores the larger pool of stationary lysosomes (see comment above).

To avoid issues with overheating, we used a short-pulsed protocol shown by Ting lab to be optimal to activate LOV-turbo. Under these experimental conditions, we did not observe any overheating of the media, morphological changes or loss of neuronal integrity i.e., fragmentation, which are suggestive of neuronal stress.

The 30 min was a compromise between good labelling efficiency needed for mass spectrometry and temporal resolution. Whilst APEX is expected to be cleaner due to the much shorter labelling period, it is characterised by higher noise and affected by cellular toxicity due to hydrogen peroxide used for activation. LOV-Turbo captures interactors early in the biosynthetic pathway (Golgi) and also some late ones (LAMTORS). Hence, we believe that LOVTurbo is a viable approach to perform proximity labelling in neurons.

Related to Fig2E-F, one would expect as well an identical staining for V5 and Lamp1/Lamp2a which is not the case for both sets of data. Particularly for the Lamp2a there is essentially no overlap with V5 and in this case the biotin label also doesn't match Lamp2a (and several strong spots for Lamp1 are not positive for biotin, nor V5). The authors should provide more convincing images that the model and experimental approach is reliably capturing biotinylation on lysosomes.

We modified this figure by adding arrows to show overlap events. For LAMP2A, we expect less colocalisation as the LAMP2 antibody used is pan-LAMP2 and the tagged LOVTurbo is only for LAMP2A.

- With respect to the interactome analyses, it should be more clearly mentioned in the text that experiments were done in biological quadruplicate (only mentioned in the legend to figure 3). The samples were run twice (batch 1 and 2): is there a logic explanation why the PCA between the two batches of the same conditions differ that much?

Data-dependent acquisition in mass spectrometry (used in this work) is inherently very

variable, hence the differences observed in the PCA.

More importantly, with respect to validation, the authors focus on common interactors for both Lamp1 and Lamp2a. Given the focus is on different lysosomal populations, I would argue that it is more relevant to validate unique interactors for each population. Why is this not done? Can the authors identify unique proteins and validate these likewise. Are there interactors known to be uniquely localized to either axons or dendrites: scrutinizing this more in validation experiments would improve the impact of the manuscript.

The number of unique proteins for LAMP2A was limited probably due to coverage. Hence, we validated common interactors to ensure we have good-quality data that can be used to support future experiments possibly using axonal interactomics. This rationale is now better explained in the Discussion.

Secondly, among the common interactors, they validate synaptic proteins, including Syt1 and Snap25. However, the imaging data only show Syt1 and Snap25 positive spots of which some are closely associated with Lamp1 and Lamp2a. At this resolution, one would expect to see clearly overlapping spots if they are co-localized on the same organelle. The data are not convincing to state that they represent biosynthetic Lamp organelles carrying synaptic proteins; the authors should include live imaging to demonstrate they co-traffic as well.

A recent study by Li *et. al.* (2024), also referenced in the manuscript, showed clear co-transport between SYT1 and LAMP1-positive organelles. Considering the evidence for co-transport was provided in an independent study and the limitations of colocalisation (discussed above), these data were moved to Supplemental Information.

Moreover, the authors should perform validation experiments in their microfluidic chamber to provide more conclusive data on axonal colocalization and co-trafficking. Of note, the authors conclude in their abstract that LAMP2a are a subpopulation of the Lamp1 organelles; but in their Discussion they also state that Lamp1 and Lamp2a label distinct organelle populations. From the interactome data one could favor the former, whereas the transport kinetics suggest the latter. They only found 8 proteins unique for Lamp2a, among which is Lamp2: does that not demonstrate it is a distinct pool? If part of Lamp1 population, and given Lamp proteins are abundant, Lamp2 should have been recovered in the Lamp1 interactome, no? Can the authors clarify the discrepancy in their conclusions? An easy experiment could be to perform co-immunostaining for both lamps.

We have now carried out co-transport experiments, as discussed above. In light of these new experiments we slightly changed our overall conclusions to underscore that while proteomic evidence suggests that LAMP2A-positive organelles form a sub-pool of LAMP1 compartments, the transport analysis suggests otherwise. Therefore, the identity of the single-labelled organelles is an important question that needs further investigation in the future, particularly given their distinct transport dynamics.

Second decision letter

MS ID#: jcs.264466R1

MS Title: LAMP1 and LAMP2A localise to axonal organelles with distinct motility dynamics and partially overlapping molecular signatures in human neurons

Authors: Reem Abouward; Alya Masoud Abdelhafid; Oscar G Wilkins; Song-Yi Lee; Fairouz Ibrahim; Mark Skehel; Alice Ting; Nicol Birsa; Jernej Ule; Giampietro Schiavo
Article Type: Research Article

Dear Dr Schiavo,

I am happy to tell you that your manuscript has been accepted for publication in Journal of Cell Science, pending standard publication integrity checks.